# Movement, drivers and bimodality of the South Asian High

Matthias Nützel[1], Martin Dameris[1], and Hella Garny[1]

[1]Deutsches Zentrum für Luft- und Raumfahrt, Institut für Physik der Atmosphäre, Oberpfaffenhofen, Germany

*Correspondence to:* Matthias Nützel (matthias.nuetzel@dlr.de)

**Abstract.** The South Asian High (SAH) is an important component of the summer monsoon system in Asia. In this study we investigate the location and drivers of the SAH at 100 hPa during the boreal summers of 1979 to 2014 on interannual, seasonal and synoptic time scales using seven reanalyses and observational data. Our comparison of the different reanalyses especially focuses on the bimodality of the SAH, i.e. the two preferred modes of the SAH centre location: the Iranian Plateau to the west and the Tibetan Plateau to the east. We find that only the National Centers for Environmental Prediction - National Center of Atmospheric Research (NCEP/NCAR) reanalysis shows a clear bimodal structure of the SAH centre distribution with respect to daily and pentad (5-day mean) data. Furthermore, the distribution of the SAH centre location is highly variable from year-to-year. As in simple model studies, which connect the SAH to heating in the tropics, we find that the mean seasonal cycle of the SAH and its centre are dominated by the expansion of convection in the South Asian region (70° E–130° E x 15° N–30° N) on the southeastern border of the SAH. A composite analysis of precipitation and outgoing longwave radiation data with respect to the location of the SAH centre reveals that a more westward (eastward) location of the SAH is related to stronger (weaker) convection and rainfall over India and weaker (stronger) precipitation over the West Pacific.

## 1 Introduction

The South Asian High (SAH) or Asian (summer) monsoon anticyclone is one of the most pronounced circulation patterns in the northern hemisphere (NH) (Mason and Anderson, 1963) and emerges through diabatic heating in the South Asian monsoon region (Gill, 1980; Hoskins and Rodwell, 1995) during boreal summer. Horizontally, the SAH covers large parts of southern Asia and the Middle East (black contours in Fig. 1). It is located on the edge of the tropics and subtropics, vertically spanning from around 300 hPa to 70 hPa (see e.g. Fig. 2 in Randel and Park, 2006), i.e. approximately the whole upper troposphere lower stratosphere (UTLS) region. Despite the closed anticyclonic flow often shown in climatological analysis, the circulation system exhibits strong variability in strength and location (Hsu and Plumb, 2000; Popovic and Plumb, 2001; Garny and Randel, 2013; Ploeger et al., 2015; Vogel et al., 2015).

Apart from the highly variable synoptic behaviour of the SAH, Zhang et al. (2002) have found that the longitudinal distribution of the SAH centre location - as identified by the geopotential height maximum along the ridge line (see green line in Fig. 1a) - is bimodal. Using pentad (5-day) mean data, they have found two preferred modes of the SAH at 100 hPa and have coined the terms Iranian Mode (IM) and Tibetan Mode (TM) according to the two peaks (at 55° E–65° E and 82.5° E–92.5° E, respectively)

of the bimodal distribution (see cyan bars in Fig. 1a).

This bimodality has been attributed to the so called warm preference of the SAH (i.e. the SAH centre is located on top of an anomalously warm air column in the troposphere see e.g. Fig. 2 in Randel and Park (2006)) and Zhang et al. (2002) argued that the TM corresponds to diabatic heating of the Tibetan Plateau (TP) and the IM to adiabatic heating in the free troposphere and diabatic heating of the Iranian Plateau (IP). This warm or heat preference is also supported by Qian et al. (2002), who focus on the seasonal variation of the SAH, referring to the high pressure system that moves to the western Pacific during winter (see also Zhou et al. (2006) and references therein).

Consequently, following studies seized the suggestion of this bimodality (e.g. Zhou et al., 2009; Zarrin et al., 2010; Yan et al., 2011), while others (Garny and Randel, 2013; Ploeger et al., 2015) tried to link their results to this finding (see details in Sect. 2.2). The classification of the SAH into two modes has also found its way into textbooks on the monsoon system (see e.g. Yanai and Wu, 2006; Xu and Zhang, 2008; Wu et al., 2008; Zhang and Zhi, 2010).

In this study we investigate the location and movement of the SAH at 100 hPa with particular focus on the bimodality by employing seven reanalyses, including four high resolution (model resolution $<1°$) reanalysis data sets. A deeper understanding of the SAH has two major impact areas: First, on the regional scale the location of the SAH was found to be connected to precipitation anomalies in Asia (flood/drought areas) and was found to be a predictor of monsoonal spills (see Zhang et al., 2002, and references therein). Second, on the global scale, as the SAH features trace gas anomalies - e.g. of CO (Li et al., 2005; Park et al., 2009), $H_2O$ (Randel and Park, 2006), HCN (Randel et al., 2010), HCFC22 (Chirkov et al., 2016) and $O_3$ (Randel and Park, 2006) - which can ultimately reach the stratosphere (Dethof et al., 1999; Randel et al., 2010). Recent studies have successfully demonstrated the impact of the Asian summer monsoon on the composition of the extratropical lower stratosphere over Europe as measured during the TACTS/ESMVal (2012) campaign (Vogel et al., 2014; Müller et al., 2016; Vogel et al., 2016). Consequently, detailed knowledge of the location and movement of the SAH is necessary to be able to understand how trace gas anomalies build up and to quantify the amount of trace gases injected into stratosphere. Because in situ measurements at UTLS levels in the Asian monsoon and neighbouring regions are sparse, aircraft measurement campaigns have been specifically designed to investigate transport processes within the anticyclone and consequent outflow from the anticyclone (OMO, 2015; StratoClim, 2016/2017).

The questions we want to address in this study are: (1) Is there bimodality of the SAH centre location at 100 hPa? (2) How can the "climatological" bimodality be connected to the movement on synoptic time scales? (3) What causes the movement of the SAH on synoptic, seasonal and climatological basis?

The remainder of this paper is structured as follows: In Sect. 2, we present the data and methods used in this study. Sect. 3 deals with question (1). Questions (2) and (3) are addressed in Sects. 4 and 5. Finally, we discuss our results in Sect. 6 and end with a conclusion (Sect. 7).

## 2 Data and method

### 2.1 Data

For our analyses we employ data from different reanalyses and observations. Our investigation focuses on the NH summer seasons during 1979–2014.

#### 2.1.1 Reanalysis data

The seven reanalyses we are using in this study are: **(1)** The NCEP/NCAR Reanalysis 1 (**NCEP1**) from the National Centers of Environmental Prediction (NCEP) and the National Center of Atmospheric Research (NCAR), **(2)** NCEP/DOE Reanalysis 2 (**NCEP2**) from NCEP and the Department of Energy (DOE), **(3)** the Climate Forecast System Reanalysis (**CFSR**) from NCEP, **(4)** the Japanese 25-year reanalysis (**JRA25**) from the Japan Meteorological Agency (JMA) and the Central Research Institute of Electric Power Industry (CRIEPI), **(5)** the 55-year reanalysis (**JRA55**) from JMA, **(6)** the ERA-Interim reanalysis (**ERA-I**) from the European Centre of Medium-range Weather Forecast (ECMWF) and **(7)** the Modern Era Retrospective-Analysis (**MERRA**) from the National Aeronautics and Space Administration (NASA). Further specifications of these reanalysis data sets are shown in Table 1, including information about the model resolution, assimilation scheme, data source and the corresponding references.

The data used in this study cover the NH summer seasons 1979 to 2014 (2010/2013 for CFSR/JRA25). Meteorological fields (geopotential height, wind and surface temperature) of all reanalysis data sets have been used with the provided resolution of 2.5° x 2.5°, except for MERRA which has been regridded from the native resolution (0.5° latitude by 0.67° longitude) to a 2.5° x 2.5° grid. The daily data used for the detection of the SAH centre have been obtained from the 6-hourly values of the reanalyses by simple averaging. Likewise, pentad mean data and monthly mean data are calculated from the daily data.

#### 2.1.2 Observational data

We use outgoing longwave radiation (OLR) as a proxy for convective activity. In this study OLR is obtained from the daily gridded interpolated OLR data from NOAA (National Oceanic and Atmospheric Administration) (Liebman and Smith, 1996). The data set has a resolution of 2.5° x 2.5° and covers the period June 1974 to December 2013.

Additionally, we use the Global Precipitation Climatology Project (GPCP) Version 2.2 combined precipitation data set (Adler et al., 2003). This data set combines satellite and rain gauge measurements to create a global precipitation field on a 2.5° x 2.5° grid with monthly temporal resolution. The time period covered is January 1979 to present. For daily precipitation data we employ Version 1.2 of the 1° x 1° daily GPCP data set (Huffman et al., 2001), which covers the period October 1996 to present.

As a measure of the Indian Summer Monsoon (ISM) strength we include the official all-India monsoon rainfall time series, i.e. the All-India Rainfall Index (AIRI), from the Indian Meteorological Department (IMD). This monthly time series covers the monsoon period (June–September) from 1901 to 2013 and was obtained from the IMD web site at

The seasonal (3 month running mean) Niño 3.4 index based on the Extended Reconstructed Sea Surface Temperature (ERSST) Version 4 data set (Huang et al., 2015; Liu et al., 2015; Huang et al., 2016) from NOAA (National Oceanic and Atmospheric Administration) with a base period of 1981–2010 used to measure the El Niño-Southern Oscillation (ENSO) was provided by

NOAA's CPC (Climate Prediction Center) from their web site: http://www.cpc.ncep.noaa.gov/data/indices/.

## 2.2   Method

To locate the centre of the SAH we use the method described by Zhang et al. (2002), which consists of two steps: First, based on daily (pentad/monthly/seasonal) data during June to August (1979–2014) at 100 hPa the ridge line is identified as the location of the minimum of the absolute zonal wind field at each longitude (see green line in Fig. 1a) in the area 15° N–45° N

x 30° E–140° E (box marked by the grey dashed dotted line in Fig. 1a). Second, along this ridge line the maximum of the daily (pentad/monthly/seasonal) geopotential field at 100 hPa is determined.

This results in a (lat, lon) coordinate, which represents the centre of the SAH for the respective day (pentad/month/season). For analyses with respect to pentad means, only data from 3 June to 31 August are used to get 18 "full" pentads per summer period. The analysis based on seasonal mean (JJA) data, results in one centre location of the SAH for every year/summer.

We note that in a first test phase identifying the centre of the SAH only via the maximum of geopotential in the area 15° N–45° N x 30° E–140° E gave comparable probability distribution functions (PDFs) of the SAH centre location. However, all analyses in this study are based on the two step method (i.e. detection of the ridge line followed by determination of the maximum geopotential height along the ridge line). We use this method as it has been applied in the majority of studies on pressure levels (see Table 2), especially the studies dealing with daily or pentad data. Table 2 summarizes the details of studies on the SAH

(centre) location regarding the variables, the methods, the time periods and the time scales used. Apart from Wei et al. (2014), who report a bimodal distribution of the seasonal mean SAH centre location in ERA40 data on 200 hPa, all pressure-level based studies in Table 2 rely on NCEP1 data. These studies comprise the 100 hPa level and report clear bimodality in the distribution of the location of the centre of the SAH during the NH summer months on various time scales. Table 2 includes two potential vorticity (PV) based analyses on isentropes, which try to draw a connection to the bimodality found on pressure

levels. Apart from focusing on isentropes, their method is distinct from the other studies in Table 2. Instead of looking for the centre of the SAH they investigate the occurence probability of the SAH at a certain grid point. The grid point is covered by the SAH if its PV value is below a certain threshold (e.g. 0.3/1.6 PV units at 360/380 K). The respective probability is then given as the fraction of time steps where this criterion is met. Furthermore, we note the studies from Zhou et al. (2006) and Zhou et al. (2009) as they also investigate AOGCMs (Atmosphere-Ocean General Circulation Models) with respect to their ability

to simulate bimodality in the SAH location on monthly basis.

As in the studies mentioned before, we focus on the 100 hPa level in our study to be consistent with these previous works. Furthermore, the 100 hPa level is close to the tropopause height in the monsoon area and hence this level is of particular interest regarding stratosphere–troposphere exchange.

## 3 Location of the SAH

### 3.1 Different distributions of the SAH centre in NCEP1 and ERA-I

To illustrate the extent of the SAH, which is defined by the strong signatures in the geopotential field, the climatological JJA geopotential height at 100 hPa for the period 1979–2014 is shown in Figs. 1a and 1b as black contours for NCEP1 and ERA-I data, respectively. For both NCEP1 and ERA-I the climatological centre of the SAH is located at approximately $60°$ E–$90°$ E and $30°$ N–$32.5°$ N. The latitudinal location is also indicated by the climatological ridge line (green line in Figs. 1a and 1b). Interestingly, NCEP1 already seems to show two centres in the climatological long-term mean. This is indicated by the squeezed contours at approximately ($30°$ N, $72.5°$ E).

The 2-dimensional frequency distribution of the SAH centre based on daily values for the JJA period 1979–2014 is depicted by the colour shading in Figs. 1a and 1b. For NCEP1 two clear maxima can be identified, which are collocated with the maxima of the geopotential height climatology, whereas for ERA-I the distribution seems to be more smoothed out over the whole centre region of the SAH.

These features can be more easily identified in the one-dimensional PDFs of the SAH centre with respect to its longitudinal location (red bars in Fig. 1). The rather sharp maxima identified by NCEP1 lie at $60°$ E and $87.5°$ E and extend about $\pm 5°$ to each side and correspond to the IM and the TM, respectively. This terminology corresponds to the location of the maxima over the Iranian and Tibetan Plateau (IP/TP) (orography close to the ridge line displayed via black shading in Fig. 1). This double-peak structure is not reproduced by ERA-Interim data, which show the highest values over the IP, but no localized peak over the TP.

Furthermore, at about $70°$ E–$80°$ E the SAH centre is scarcely found in NCEP1 data. This is not the case for ERA-I data, which show a small peak in this region. Figs. 2a and 2b display the probabilities of the SAH centre to be located in the IM, mid (region between IM and TM) and TM region as diagnosed via daily and pentad data (blue/pink bars for ERA-I/NCEP1) to highlight this difference.

A scatter plot of the daily location of the SAH centre as diagnosed by ERA-I versus NCEP1 (Fig. 3) shows where these differences come from. A notable number of samples falls into the TP region for NCEP1 whereas for ERA-I these samples are spread out over the region $\sim 40°$ E–$100°$ E (horizontal line in Fig. 3). Conversely, samples that are found in the mid region for ERA-I are spread out over $\sim 60°$ E–$90°$ E in NCEP1 (vertical line in Fig. 3).

To verify that the results of our analysis do not depend solely on the time period chosen, we have also calculated the one-dimensional PDFs for July-August 1980–1994 (same period as used by Zhang et al. (2002) for the pentad analysis, see Table 2). These PDFs are displayed as cyan bars in Figs. 1a and 1b and confirm that the general features of the PDFs during the two periods are in qualitative agreement. Moreover, the one-dimensional distributions in Fig. 1a resemble the distributions based on pentad mean data in Zhang et al. (2002) daily data in Yan et al. (2011) (see details of these studies in Table 2).

The obvious discrepancies between NCEP1 and ERA-I lead to the questions, where these differences come from and which reanalysis is correct. Possible reasons for the differences will be stated in the discussion (Sect. 6). However, due to the complexity of the reanalysis models and the subsequent data assimilation, we find it difficult to address these questions directly.

Hence, we use a set of seven reanalyses to determine the range of results and to see if the SAH exhibits bimodality in the sense of two pronounced (centre) regions over the IP and TP.

## 3.2 Coherent analysis of the SAH centre location in seven reanalyses

To get a more reliable answer to the question, whether there is bimodality in the location of the SAH, we employ seven reanalysis data sets in a consistent manner. In the following we will show results based on daily, pentad, monthly and seasonal mean data. For all of these time scales bimodality was found in previous studies (see Table 2).

### 3.2.1 Climatology of the SAH at 100 hPa

Figure 4 shows the 1979–2014 (1979–2010/1979–2013 for CFSR/JRA25) climatology of geopotential height during NH summer (JJA) for each of the seven reanalysis data sets together with the mean ridge line (green dashed line). All reanalyses show the climatological centre of the SAH in the region ∼50° E–95° E and ∼25° N–35° N and a mean ridge line located at approximately 30° N, with slight differences depending on the reanalysis.

MERRA and NCEP2 predict the centre region farther to the west than the other reanalyses (Figs. 4c and d). In comparison with ERA-I, JRA25, JRA55 and MERRA, the NCEP-reanalyses (Figs. 4a and d) show slightly higher geopotential height values, whereas CFSR (Fig. 4g) shows slightly lower geopotential height values. This is not attributable to the different base period for the climatology of CFSR (1979–2010). Differences between ERA-I, JRA25 and JRA55 with respect to the climatological representation of the SAH (Figs. 4b, e and f) are hardly discernible.

### 3.2.2 SAH location based on daily mean data

Figure 5 shows the location of the SAH based on daily JJA data for the period 1979–2014 (1979–2010/1979–2013 for CFSR/JRA25) for these seven reanalyses. Obviously, only NCEP1 shows a clear double-peak structure. Note that NCEP2, which has the same native resolution as NCEP1, but includes updated physics and corrections of errors (Kanamitsu et al., 2002), shows a smoothed out TM. Nevertheless, NCEP2 is the reanalysis that agrees the best with NCEP1 in terms of producing two modes. As an example, only NCEP1 and NCEP2 show a sharp peak over the IP. The reanalyses that show the best agreement are CFSR, ERA-I and JRA55. These reanalyses have a high horizontal model resolution (<1°) and ERA-I and JRA55 use 4D–Var data assimilation (see Table 1). Allthough MERRA - also a high resolution reanalysis - does not agree in detail with CFSR, ERA-I and JRA55, the following points are supported by all four of these reanalyses: the IP seems at least as important as the TP, the SAH is almost as likely located in the region 70° E–80° E as in any other region of the broad centre region (∼50° E-100° E, depending on the reanalysis), the peak over the TP is shifted farther eastward than in the NCEP1 data. We have found similar results for the analysis based on pentad mean data (cf. Fig. 2b).

When comparing the distributions of the daily location of the SAH centre for individual years, we have found strong inter-

annual variability (not shown). Despite this interannual variability, peaks in the distribution of individual years, as diagnosed by NCEP1 and NCEP2, usually are consistent with the multiannual mean displayed in Fig. 5. In contrast, the other reanalyses exhibit more variability, e.g. there are years that show a clear preference of the eastern or western side and some years also exhibit two centres of activity, however with varying geographical position.

### 3.2.3 SAH location based on monthly mean data

Figure 6 shows the PDF of the SAH centre location based on the diagnosis of monthly mean data for JJA 1979–2014 (1979–2010/1979–2013 for CFSR/JRA25). The distribution has been smoothed by taking the running average over three grid points. We include this step as for monthly (seasonal) mean data only 108 (36) data points for the years 1979 to 2014 in combination with the bin size of 2.5° result in a low ratio of data points per bin and thus (maybe) artificial peaks, which should not be overinterpreted. Moreover, sometimes the location of the maximum in geopotential is not unique, i.e. two or more neighbouring boxes have the same geopotential value.

For the monthly mean data all reanalyses show a bimodal structure with one local maximum close to 60° E and a second maximum close to 85° E (90° E for CFSR and 75° E for MERRA). The best agreement can be found between JRA55 and ERA-I. Analysing the months June, July and August separately shows that in JRA55 and ERA-I this structure is due to the distributions during June and July. In both months a double-peak structure with a notably stronger (weaker) TM (IM) in June than in July can be observed, whereas in August the distribution is rather smooth for JRA55 and ERA-I (not shown). For each of the months June, July and August, the distribution of the SAH centre based on NCEP1 (NCEP2) shows the typical bimodal structure with a more pronounced TM (IM) than IM (TM). Common to all reanalyses is that there is a shift of the distribution to the west from June to July and a shift back to the east from July to August.

### 3.2.4 SAH location based on seasonal mean data

Based on seasonal mean (JJA-mean) data the SAH shows a bimodal structure in the reanalyses NCEP1 and NCEP2 (see Fig. 7). Here, NCEP1 shows a pronounced peak over the TP and a second one over the IP, whereas NCEP2 shows only a sharp peak over the IP. Additionally, JRA25 shows low probabilities around 70° E. In contrast, CFSR, ERA-I, JRA55 and MERRA show high probabilities over the whole centre region (∼60° E–85° E, depending on the reanalysis).

Our analysis of the SAH centre location at 100 hPa indicates that based on daily, pentad and seasonal data only NCEP1 shows a clear bimodal structure. We emphasise that in particular the results based on daily and pentad data are of interest as they should be linked to the synoptic movement of the SAH. Based on monthly mean data all reanalyses show higher probabilities of occurrence over the TP and IP. The occurrence of the SAH centre based on daily, pentad, monthly and seasonal mean data is summarised in Fig. 2. Different probability distributions of the SAH with respect to daily, pentad, monthly and seasonal mean data arise as there is no weighting of the strength of the SAH centre with respect to its surrounding. This issue is addressed with a more visual explanation in the next section. We also note that the quantitative results are likely to depend

on the height level and time period chosen.

The salient disagreement of the reanalyses in the distribution of the SAH center location is our motivation to revisit the questions of how the SAH moves on various time scales and how this movement is caused. To tackle these questions, we will focus on results based on observational and ERA-I data during the next two sections (Sects. 4 and 5). We choose ERA-I as it is a heavily used reanalysis with the most recent data assimilation scheme. Apart from that, our choice is arbitrary and we address the sensitivity of the presented results with respect to the reanalysis in the discussion (Sect. 6).

## 4   Movement and drivers of the SAH

Simple model studies have shown that constant diabatic heating in South Asia causes a mean UTLS circulation to its northwest, which resembles the climatological SAH (see e.g. Gill, 1980; Hoskins and Rodwell, 1995). As diabatic heating in the southern monsoon region is largely caused by the latent heat release due to convection, we use OLR as a proxy for convective activity and consequently for diabatic heating.

Figure 8 shows the temporal evolution of ERA-I geopotential (averaged over $20°$ N–$40°$ N) during the summer months of 1983 and 1987. Choosing these two years is arbitrary, however they are useful to illustrate common and individual features of the monsoon season. The green lines indicate the location of the SAH centre as diagnosed via the method described in Sect. 2.2 based on daily (light green) and pentad (dark green) data from ERA-I. Additionally, we have included mean OLR in the region $15°$ N–$30°$ N (main convective region south of the SAH) from NOAA at the levels 190 W m$^{-2}$ and 180 W m$^{-2}$ (black contours).

In Fig. 8 the lowest OLR values are mostly confined to the area $75°$ E–$105°$ E and are mostly located east of the highest geopotential height values. East of the OLR minimum we can observe eastward migration of high geopotential, associated with eastward eddy shedding of the anticyclone. A strong shedding event is observed in mid August 1983 (turquoise star in Fig. 8a). West of the OLR minimum region, the core of the anticyclone usually propagates westwards. Another feature visible in the Hovmoeller diagrams are splittings of the anticyclone, e.g. $\sim$10 July 1983, $\sim$10 July 1987 and end of July 1987 (indicated by arrows in Fig. 8). Splitting events and the development of high geopotential values close to the OLR forcing are often a cause for "jumps" of the location of the SAH centre (dark green line in Fig. 8).

Figure 9 shows geopotential height along the ridge line during June to August 1983 for individual pentads, monthly mean and seasonal mean data. We note that in 1983, based on daily means, all reanalysis show a distribution of the SAH centre that has one strong centre at approximately $60°$ E and a weaker maximum at approximately $95°$ E (not shown). The SAH centre is located over the TP in June and over the IP in July and August (coloured dashed lines in Fig. 9 and purple dots in Fig. 8a). The seasonal mean shows a maximum over the IP (dashed black line in Fig. 9). The effect, which leads to different distributions with respect to varying time scales, can be inferred from the ridge lines for the months June and August in Fig. 9: During June the maximum in the TP is weak, whereas during August the maximum over the IP is pronounced. Nevertheless, both months contribute equally to the distribution of the SAH based on monthly mean data, whereas based on seasonal mean data, the peak

will be detected over the IP only. In 1987 the east mode is found based on seasonal data and a rather smoothed out distribution is found based on the daily analysis.

To illustrate the climatological connection of OLR and the SAH we display the JJA climatology (1979–2013, i.e. the overlapping time period of NOAA-OLR and ERA-I data) of OLR in Fig. 10 (orange contours). Additionally, we show the climatology

of vertical velocities at 100 hPa as diagnosed from ERA-I (colour-shading) and the low level winds (grey vectors), e.g. identifying the Somali-Jet, which brings moisture from the Arabian Sea to India (Rodwell and Hoskins, 1995).

The deep convective region is located to the southeast of the climatological location of the SAH (dashed black contour Fig. 10), with the lowest OLR values (below 180–190 W m$^{-2}$) over the Bay of Bengal. Upward (downward) winds are located on the eastern (western) side of the SAH in agreement with Rodwell and Hoskins (1996).

The mean seasonal evolution of the SAH location and strength together with OLR during May–September (1979–2013) is shown in Figs. 11 a and 12. Until approximately mid July the area of strong convective activity extends northwestwards and retreats southeastwards later. In a similar way the location of the SAH moves northwestwards during the build up of the SAH and southeastwards during the decay phase of the SAH (shifting ∼30° longitudinally and ∼10°–15° latitudinally). The seasonal east–west shift can be also found in daily precipitation data from GPCP during the period 1997–2013 (see Fig. 11 b) and the

seasonal northward migration of precipitation has been noted in previous studies (e.g. Yihui and Chan, 2005, their Figure 3). We note that in contrast to the expansion of low OLR and high precipitation values, the region of lowest OLR and highest precipitation (∼90° E) in Fig. 11 does not shift notably.

To further study the relation of convection (OLR) with the SAH we investigate the temporal correlation of NOAA-OLR with geopotential at 100 hPa from ERA-I on subseasonal time scales. Therefore, we calculate time lag correlations of OLR aver-

20 aged over the region 70° E–130° E x 15° N–30° N (i.e. the deep convective region on the southeast border of the SAH) with geopotential averaged over 20° N–40° N. Before calculating the correlations based on data from May–September 1979–2013, the data were deseasonalised and oscillations with a period of less than 10 days were removed. The results of the time lag analysis are shown in Fig. 13. At around time lag 0 the maximum anticorrelation ($< -0.50$) is found at approximately 75° E–85° E and moves westward with increasing time lag. Approximately 3 days later the maximum anticorrelation ($< -0.45$) is found

around 45° E. East of the instantaneous response region the maximum anticorrelation travels eastward more slowly, e.g. the maximum anticorrelation at a time lag of 4 days ($< -0.35$) can be found at approximately 90° E–95° E.

The northwest–southeast movement found in the seasonal cycle of the SAH can also be identified on the interannual time scale. Table 3 shows the correlation of longitude and latitude occurrence of the SAH centre for the seven reanalysis data sets. The correlation coefficients are calculated based on the seasonal mean and monthly mean data (1979–2013/2014). For the latter the

30 multiannual mean of each month has been subtracted in order to deseasonalise the data (in the following this will be referred to as deseasonalised monthly mean data).

For seasonal mean and deseasonalised monthly mean data all reanalyses show that westward (eastward) movement of the SAH is related to northward (southward) movement. The separate analysis of June, July and August yields that this relationship is strong during June and July (significant on the 10% level in all reanalyses). In August, however, the connection is weaker and

35 gets insignificant for most reanalyses (on the 10% significance level, weakest anticorrelation of -0.08 found in MERRA data).

In summary, we have found that the SAH's location and strength is notably related to the location and strength of convection located on its southeastern border (on climatological, seasonal and subseasonal time scales). This connection is especially prominent in the mean seasonal evolution. Moreover, the seasonal northwest–southeast movement of the SAH is also evident in the seasonal mean and in the deseasonalised monthly mean data in summer, leading to the hypothesis that changes in the location of convection are related to the movement of the SAH on these time scales as well. This hypothesis will be tested via composite analyses in the next section (Sect. 5).

## 5 Composite analyses of west and east phase

Regardless of the existence of two preferred spatial modes of the SAH, it is of great interest to identify signatures that are associated with an eastward or westward location of the SAH centre. In the following the days, months and summers/years with a rather (see exact definition later) westward (eastward) location of the SAH centre, will be termed west (east) days, months and summers/years.

A similar method has been applied by Yan et al. (2011), who analysed satellite measurements (of $O_3$, $H_2O$ and CO) with respect to the location of the SAH as diagnosed by NCEP1 daily data. These composites show a dipole pattern in the distribution of trace gas anomalies where positive (negative) values of tropospheric (stratospheric) tracers are collocated with the current location of the SAH. This illustrates how trace gas anomalies follow the movement of the SAH (see also Randel and Park, 2006; Garny and Randel, 2013).

The importance of the location of the SAH centre can be inferred from Fig. 14, which displays geopotential height composites (grey contours) at $100\,\mathrm{hPa}$ together with anomalies of the vertical velocities at $100\,\mathrm{hPa}$ for west (Fig. 14a) and east (Fig. 14b) summers during 1979–2013. A year belongs to the west (east) composite if the seasonal SAH centre is located more than $7.5°$ to the west (east) of the multiannual mean location of the SAH centre (resulting in 8 summers for each composite). During west years the SAH tends to be slightly stronger and negative anomalies in the vertical velocities (i.e. relative upward transport in the order of 25% of the maximum climatological values cf. Fig. 10) in the centre region are found. This could be an indicator for stronger confinement and enhanced upward transport during west summers. For the eastern summers, the anomalies are exactly reversed, i.e. regions of anomalous upward motion during west years show anomalous downward motion during east years and vice versa.

To identify the signatures associated with the longitudinal location of the SAH, we will use composite differences of OLR (NOAA) and precipitation (GPCP). In detail, the respective data are split according to the location of the SAH centre in ERA-I data into a west and an east composite. Finally, the two composites are subtracted from each other (we show results as: west minus east).

We will present analyses based on seasonal mean, monthly mean and daily data during June to August. To separate the effect of the seasonal cycle (see Figs. 11 and 12) from subseasonal processes within the monthly mean and daily data, we use the deseasonalised monthly and daily means of OLR, precipitation and surface temperature (for daily data the June to August period of the smoothed seasonal cycle based on May to September data from 1979–2013 has been removed). Accordingly, the split

into east and west phase is done with respect to $\pm 7.5°$ deviation from the multiannual summer mean, from the multiannual monthly means or from the smoothed seasonal cycle (see dashed line in Fig. 11) of the longitudinal position of the SAH centre. The following results are also supported (qualitatively) if we split according to $\pm 1\sigma$, where $\sigma$ represents the multiannual seasonal, multiannual monthly or multiannual daily standard deviation. Similarly, splitting the data into west ($<67.5°$ E) and east ($>80°$ E) phase or location over the IP and TP (both plateaus as defined before) leads to comparable results. This might be due to the fact that subseasonal variations dominate the seasonal variations. For the seasonal data the three methods give almost the same composites (see Fig. 2d, mean location of 73.5° and standard deviation of $\sim 9°$) and hence similar results.

Figure 15 shows composite differences of OLR and precipitation with respect to different time scales (for daily data there is no precipitation composite as daily resolved GPCP data is only available since October 1996). For seasonal, monthly and daily data the west (east) composite is comprised of 8, 39, 1222 (8, 38, 1087) data points, respectively. Areas that do not reach the 10% significance level are dotted.

Two main regions that show significant differences between west and east phases in OLR and in precipitation are India and the western Pacific. In detail, OLR is lower (higher) during west (east) periods over India and the Arabian Sea. For the western Pacific the reverse connection is found. Differences in precipitation are accordingly, i.e. lower (higher) OLR values are accompanied by more (less) rainfall. Additionally, there is less OLR in the deep tropics, indicating more convective rainfall in this region during western phases. This is most pronounced in the monthly data (Figs. 15c and 15d). In comparison with seasonal and monthly mean data (Figs. 15a and 15c) there is an important difference for the daily data (Fig. 15e), as negative OLR values stretch farther from west to east at approximately 20° N–30° N.

The analysis for June, July and August separately (not shown) shows that in June and July, the significant differences are mostly in agreement with the results found for the monthly JJA data. In August, however, almost no significant differences of precipitation can be found over India.

In short, we have found that the west–east location of the SAH is connected to opposing anomalies of convection and precipitation over India and over the West Pacific with respect to daily, monthly and seasonal data. Stronger (weaker) precipitation over India (the West Pacific) is related to a more westward location of the SAH centre.

# 6 Discussion

The comparison of the daily location of the SAH centre during JJA 1979–2014 as diagnosed from NCEP1 and ERA-I shows that NCEP1 exhibits strong bimodality in its longitudinal location (in agreement with Zhang et al., 2002; Yan et al., 2011), whereas ERA-I shows only a pronounced signature over the IP. This difference is also visible in the long-term climatology, i.e. there is bimodality in the NCEP1 climatology of geopotential at 100 hPa.

In the analysis of seven reanalysis data sets (CFSR, ERA-I, JRA25, JRA55, MERRA, NCEP1, NCEP2) with respect to the location of the SAH we find that only NCEP1 produces a pronounced maximum over the TP and a distinct minimum in the region 67.5° E–80° E, i.e. between the IP and the TP. Furthermore, only NCEP1 and NCEP2 show a sharp peak over the IP. Although there are differences between all reanalyses, NCEP2 and especially NCEP1 are outliers regarding the distribution of

the SAH centre at 100 hPa. The reanalysis data sets that show the best agreement regarding the location of the SAH centre are CFSR, ERA-I and JRA55.

The analysis of individual years shows strong interannual variability in the location of the SAH. This variability limits the application of the findings for the long-term mean to single years and vice versa. E.g. the distribution of low PV at 380 K in 2006 as shown in Garny and Randel (2013) (see their Fig. 15), which is based on MERRA data, exhibits high values on the western side ($\sim$30° E–70° E). This is in good agreement with the distribution of the SAH centre location at 100 hPa in 2006, which is rather shifted to the west in all reanalyses, except for NCEP1 (not shown).

Based on monthly mean data over the period JJA 1979–2014 (2010/2013 for CFSR/JRA25) we have found that all reanalyses (except for MERRA) show two regions of increased probability, which lie over the IP and TP (see Fig. 2c). However, as for the daily data, based on the monthly data NCEP1 (NCEP2) shows a more pronounced TM (IM) than ERA-I, JRA55 and JRA25. After analysing the months June, July and August separately we have found that in the latter three reanalyses two centres of activity can be found in June and July (weaker IM in June and stronger IM in July), whereas in August the distribution is rather smooth for ERA-I and JRA55 (not shown).

Based on seasonal data we can identify bimodality in NCEP1 and NCEP2. As before NCEP1 (NCEP2) shows a strong TM (IM) and a weaker IM (TM). Additionally, JRA25 shows low occurrences of the SAH in the region between the IP and TP. CFSR, ERA-I, JRA55 and MERRA show a spread out distribution over the region 55° E–90° E with a single peak at $\sim$70° E–80° E, depending on the reanalysis.

Possible reasons for the different distributions as given by the reanalyses with respect to varying time scales (except for NCEP1 and NCEP2, which show consistent distributions on all time scales) are as follows: (1) The method of locating the SAH centre picks the highest local maximum for individual samples and does not take the relative strength of the current SAH centre into account. (2) It is rather unlikely that the SAH centre is located west of 55° E or east of 92.5° E for many subsequent days, hence there is almost no occurrence of the SAH in this region based on monthly means. However, on a daily/pentad basis locations to the west of the IP and to the east of the TP have a significant weight (see Fig. 2), thus putting weight to the IM or TM on a monthly basis when the SAH resides rather to the west or east. (3) The bimodality in the monthly data might be influenced by the seasonal cycle.

The reasons for the differences in the SAH centre location between the reanalyses are attributable to the underlying model, the assimilation technique and the observational data, which are assimilated by the different reanalyses. While it is not possible for us to disentangle the relative influence of these sources some hints might be as follows: Numerous changes have been made from NCEP1 to NCEP2. These changes affect the thermal and orographic forcing of the IP and TP as well as the diabatic heating associated to tropical convection e.g. through changed boundary conditions, updated physics and smoothed orography (Kanamitsu et al., 2002). These changes have a large impact on the distribution of the SAH centre location (see e.g. Fig. 5). This emphasises the impact of model properties on the SAH centre location. Since some of the features described above are only shown by NCEP1 and NCEP2 the importance of the data assimilation scheme (e.g. inclusion of TOVS and ATOVS temperature profiles for NCEP1 and NCEP2 vs. radiance data directly for the other reanalyses) might be inferred. However, this could

also be related to the closeness of the underlying model. During the time period 2004–2013, when advanced observational data, which were not included in NCEP1 and NCEP2, are available, similar distributions for daily and pentad data are found. Together with the distribution presented in Fig. 1 for the period 1980–1994, this indicates that the time period chosen does not influence our results qualitatively. This in turn brings us to the hypothesis that the transition between different observational

data sets, which are included in the reanalyses are of minor importance.

Wright and Fueglistaler (2013) have found large differences in the climatologies of diabatic heating rates among different reanalyses. These diabatic heating rate differences (and connected differences on shorter time scales) can be expected to have an impact on the distributions of the SAH centre location (on daily and pentad basis) with respect to the various reanalyses. The most prominent difference in the distribution of the SAH centre location is that the clear bimodality on short time scales found

in previous studies (mostly based on NCEP1 data) is not visible in most recent reanalyses. We cannot answer the question which reanalysis represents the reality the best. Nevertheless, the fact that modern reanalyses do not produce the bimodality of the SAH centre location with respect to daily, pentad and seasonal data strongly suggest that the bimodality found on these time scales using NCEP1 data is an artefact of this particular reanalysis.

Previous studies which address the bimodality of the SAH have mostly focused on the 100 hPa level (see Table 2). To see

how robust our results are, we employed ERA-I on the 395 K level. The SAH centre location was defined as the maximum of the Montgomery streamfunction along the ridge line. We found that the PDFs of the SAH centre location with respect to daily and monthly data are similar to ERA-I on 100 hPa. For the seasonal mean data we have found that the distribution changes in favour of the TM and IM, i.e. for seasonal data 12, 10, 14 years are located in the IP, mid and TP region, respectively.

To assess what drives the variability of the SAH we show the movement of the SAH centre and the temporal evolution of geopotential at 100 hPa from ERA-I. We find that geopotential often moves to the west and that less regularly shedding occurs to the east of the SAH centre. The mean seasonal evolution of convection (in the South Asian tropical region, here 70° E–130° E x 15° N–30° N ) and mean seasonal location of the SAH centre as diagnosed by ERA-I show a clear connection (see Figs. 11 and 12): As the region of low OLR and strong precipitation (deep convective region) extends northwestwards during

the build up phase of the SAH, the SAH and its centre move northwestwards as well. Once the region of strong convection withdraws southeastwards, the SAH centre follows accordingly. This is in agreement with Gill (1980) and the climatologies of geopotential height and OLR during JJA shown in Figs. 4 and 10 and has been discussed based on monthly data for the retreat phase of the SAH in Lau et al. (1988). The smoothed mean seasonal movement of the SAH as diagnosed from NCEP1 behaves accordingly, however with a slightly longitudinal range of approximately ∼25° compared to ∼30° for ERA-I.

A time lag analysis, linking convective activity in the tropics and the evolution of geopotential at 100 hPa on subseasonal time scales, shows that the instantaneous response of the geopotential field to convective activity is located on the western edge of the forcing region, again in agreement with results of Gill (1980). Overall, the evolution of geopotential and its connection to OLR (convection) is in agreement with findings based on MERRA data from Garny and Randel (2013), who link divergence associated to deep convection with the evolution of the area of low potential vorticity (PV) at 360 K (see their Figs. 6 and 8).

When performing the time lag analysis with an averaging area of OLR, which extends farther to the north, we found weaker

anticorrelations. Furthermore, when averaging OLR over the TP (approximately: 70° E–105° E x 30° N–40° N) only, we get positive correlations of OLR with geopotential height located over the TP (in contrast to negative correlations which associate reduced OLR with increased convection and strengthening of the SAH). Two possible reasons for this are: (1) The SAH is not powered but maybe fed by the convection over the TP with respect to trace gases (Heath and Fuelberg, 2014). (2) (Low) OLR in this region is not a reliable measure for convection due to the height of the TP. This might be enhanced by sampling biases due to the sun-synchronous orbit of the NOAA satellites. These biases are more important over land as convection has a stronger diurnal cycle over land than over sea (Liu and Zipser, 2008).

Composites of east and west summers during 1979–2013 and corresponding anomalies of the vertical velocities at 100 hPa indicate the possible connection of a more eastward or westward location of the SAH with altered stratosphere–troposphere exchange in the monsoon region. The analysis of composite differences of OLR and precipitation for west and east summers/months, based on the location of the SAH centre, yields anomalies of convection and precipitation between these summers/months. There is notably more convection over the Arabian Sea (and India) and consequently more precipitation over India when the SAH centre is located more westward. In contrast, during eastern summers/months there is more convective activity and stronger precipitation over the West Pacific. Additionally, during the west summers/months negative surface temperature anomalies can be found over India, probably connected to stronger precipitation. Furthermore, in the monthly analysis we have found that these signatures mostly come from the months June and July.

As the composite analysis of convection and precipitation with respect to seasonal mean and monthly mean data suggests a connection of the SAH location and the ISM (measured by IMD's AIRI), we display the relationship of the seasonal SAH centre location (from ERA-I) and the seasonal AIRI in Fig. 16a. Figure 16b shows the correlation coefficients of the location of the SAH centre with the AIRI over the time period June to August 1979–2013 for the seven reanalyses (1979–2010 for CFSR). For the calculation of the monthly mean correlation coefficients the data was deseasonalised (i.e. multiannual monthly means have been subtracted from the SAH location and the AIRI time series).

All reanalyses show stronger anticorrelations based on seasonal mean than on monthly mean data, except for CFSR and NCEP2. The latter does not produce a significant anticorrelation on the seasonal time scale. Moreover, for the separate months the connection between the SAH location and the ISM in June and July is stronger than in August (except for JRA25, June -0.32 vs. -0.33 in August) and gets insignificant – on the 10% level – for CFSR, JRA55, MERRA, NCEP1 and NCEP2. This is in agreement with the findings from the composite analysis.

The Asian monsoon system and especially the ISM is influenced strongly by ENSO (see e.g. Mooley and Parthasarathy, 1983). Hence we analysed the impact of ENSO on the connection between seasonal SAH center location and ISM by calculating partial correlations between the SAH's longitudinal location from ERA-I and the AIRI index with respect to the seasonal Niño 3.4 index from CPC. Partial correlations were calculated with respect to the lagged seasonal Niño 3.4 starting with DJF before the summer monsoon and ending with NDJ following the monsoon period. These partial correlations were ranging from -0.44 to -0.51, i.e. comparable to the simple correlation coefficient between the two indices. The lagged correlations between the SAH's location and Niño 3.4 were not significant and below 0.1.

The relationship between the ISM and the SAH is also supported by findings of Wei et al. (2014, 2015), who linked the west–

east and northwest–southeast displacement of the SAH on interannual time scales with the strength of the ISM. In detail they used two indices the $SAHI_{14}$ and the $SAHI_{15}$ (see caption of Fig. 14) - as given by geopotential from ERA40 on 200 hPa (based on JJA data during 1958–2002) - and found correlation coefficients of -0.49 for the $SAHI_{14}$ and -0.64 for the $SAHI_{15}$ with the rainfall over India (note that a more positive SAHI corresponds to a more eastward location of the SAH). We calculated

the correlation of the $SAHI_{14}$ and the $SAHI_{15}$ for ERA-I geopotential at 100 hPa and AIRI from the IMD as -0.57 and -0.61, respectively. Additionally, Wei et al. (2014) have performed idealised model studies and have indeed shown the importance of latent heat release over India to the location of the SAH on interannual time scales. Wei et al. (2014) also address the possible influence of ENSO on the SAH's longitudinal location and come to a similar conclusion that ENSO does not play a major role for west–east shifts of the SAH on interannual time scales.

However, in contrast to the finding of Wei et al. (2015), who identify an opposing heating/cooling source over the Yangtze river valley using GPCC data from 1958–2002, we do not find a significant connection of rainfall in this region with the location of the SAH, but rather with convective activity over the West Pacific. This excess/deficit precipitation over the West Pacific was also found in some of the CGCM based precipitation composites, which were presented in Zhou et al. (2006) (their Fig. 4).

The composite difference of OLR based on daily data shows similar results as for the seasonal and monthly data. To check

how robust the signatures are, we have built composites based on the SAH centre location at 100 hPa from NCEP1. In general, the results match those, when the splitting is performed based on ERA-I. Especially for the daily (1344 west; 1305 east) and monthly (35 west; 39 east) mean data the results match well. For the seasonal data (9 west; 10 east) the signatures in OLR and precipitation over India are shifted farther towards the southern slopes of the Himalaya.

As plateau heating has been discussed as a driver for the SAH to shift (e.g. Zhang et al., 2002; Liu et al., 2007), we have

performed the composite analysis on reanalysis surface temperature data for ERA-I and NCEP1 (not shown). The results show that during western periods surface temperatures are usually lower (up to -1.25 K) over India, likely associated with heavier rainfall in this region. With respect to the IP (TP), positive (negative) anomalies in surface temperatures get more pronounced on shorter time scales. This is especially the case for ERA-I data, where only the composite based on daily data shows a clear and significant negative anomaly over the TP during west phases, whereas for NCEP1 data the TP and IP anomalies are strong

regardless of the time scale. This might be hinting that the importance of heating associated with the two plateaus is of more importance on shorter time scales and in NCEP1 compared to ERA-I data.

The drawback of using reanalysis surface temperature data, especially in the TP (and probably IP) region where observational data are scarce, is that this variable is strongly influenced by the model itself (at least for NCEP1 see Kalnay et al., 1996). Hence the signatures in surface temperatures over the TP might not be reliable and thus rather reflect the models' connection

of surface temperature over the TP with the location of the SAH.

## 7  Conclusion

In this study the movement and drivers of the SAH during the period 1979–2014 are investigated using observational and reanalysis data. Special attention is brought to the subject of bimodality, i.e. the two preferred modes of the SAH's centre location over the Tibetan and Iranian Plateau.

We find that bimodality with respect to daily, pentad and seasonal data at 100 hPa is only found in NCEP1 and NCEP2, however not consistent to each other. Although we cannot rule out that NCEP1 or NCEP2 simulate the distribution of the SAH centre correctly, the other reanalyses – including most recent ones (e.g. ERA-I and JRA55) – do not support the notion of bimodality as two designated centres of activity. This is of special interest as a couple of studies have conducted analyses based on this concept. Thus it might be useful to investigate if their findings are affected when more recent reanalyses are being used. Further, it might limit the conclusions drawn from these studies, i.e. the drivers associated with the two modes might rather reflect the model properties than the actual atmospheric situation and thus might vary from reanalysis to reanalysis. Finally, using a more recent reanalysis might enhance results found in composite differences, e.g. with respect to satellite measured trace gases as in Yan et al. (2011).

With respect to the drivers of the SAH, we find that shifts in convection are a main cause for the shift in the location of the SAH (on various scales): We connected the mean seasonal evolution of the SAH to the seasonal cycle of convection in the tropical region adjacent to the SAH. A modified extension of the composite analysis performed in Wei et al. (2014, 2015) showed that the ISM and convection over the West Pacific are related to the longitudinal position of the SAH centre. Hence the location of the SAH might be related to different boundary layer source regions and in turn affect consequent transport (maybe into the stratosphere, cf. Fig. 14).

We note that on top of the influence through the location of convection, internal variability – i.e. instability of the anticyclone and subsequent westward movement or splitting (Hsu and Plumb, 2000; Popovic and Plumb, 2001) – plus external forcing (described by Dethof et al., 1999) influence the location of the SAH on the synoptic time scales. Additional influences from the orography and heating of the TP and IP might also modulate the location of the SAH (e.g. Zhang et al., 2002; Liu et al., 2007, and references therein). The relative importance of the thermal forcing of the TP and convection in the South Asian region on the synoptic movement of the SAH is still an open point and difficult to evaluate from reanalysis and observational data only.

*Acknowledgements.*  We thank William Randel, Klaus-Dirk Gottschaldt and Helmut Ziereis for their helpful comments on the manuscript. We acknowledge the use of CDO (Climate Data Operators) for data analysis and data processing. This software is available at http://www.mpimet.mpg.de/cdo. We used the NCAR Command Language (NCL) for data analysis and to create the figures of this study. NCL is developed by UCAR/NCAR/CISL/TDD and available on-line: http://dx.doi.org/10.5065/D6WD3XH5. We acknowledge the use of interpolated OLR and GPCP V2.2 monthly precipitation data provided by the NOAA/OAR/ESRL PSD, Boulder, Colorado, USA, from their Web site at http://www.esrl.noaa.gov/psd/. GPCP V1.2 daily precipitation data was provided by NCAR's Research Data Archive (http://www.rda.ucar.edu). We acknowledge the institutions listed in Table 1 for the production and dissemination of reanalysis data. This work was funded by the EU project StratoClim (grant number StratoClim-603557-FP7-ENV.2013.6.1-2).

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

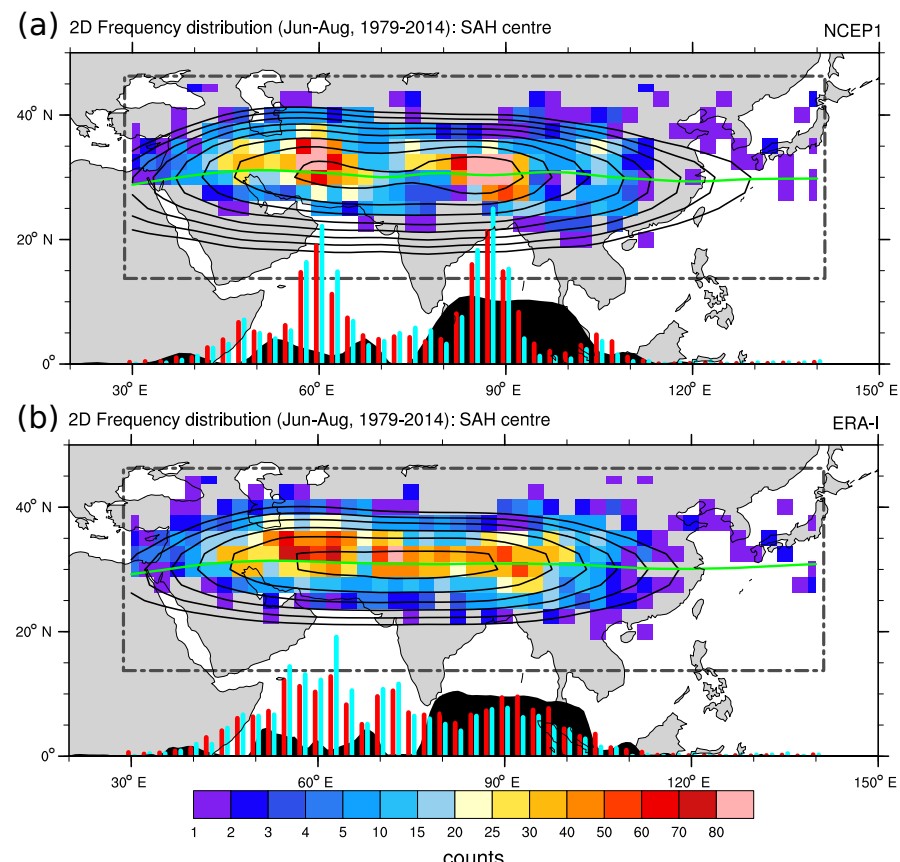

**Figure 1. (a-b)** Colour shading indicates the two-dimensional frequency of occurrence of the SAH centre at 100 hPa as diagnosed by **(a)** NCEP1 and **(b)** ERA-I over the June to August 1979–2014 based on daily values (2.5° x 2.5° bins; note the nonlinear colour scale). The box marked by the grey dashed dotted line indicates the range of the data which is used to diagnose the centre. Black contours show the long-term seasonal (JJA, 1979–2014) mean of the geopotential height (contour levels starting at 16.72 km and spacing of 15 m) and the green line shows the long-term mean location of the ridge line (zero zonal wind) at 100 hPa. On the longitude-axis, black shading indicates **(a)** NCEP1 orography (at T62 resolution) and **(b)** ERA-I orography (0.75° resolution ∼ ERA-I native resolution) at ∼ 31.5° N (approximately the ridge line), 1 degree corresponding to 500 meters. Red (cyan) bars indicate the one-dimensional PDF (bins of 2.5°) of the daily location of the SAH centre over the June–August (July–August) period 1979–2014 (1980–1994) with 2 degrees corresponding to 1%.

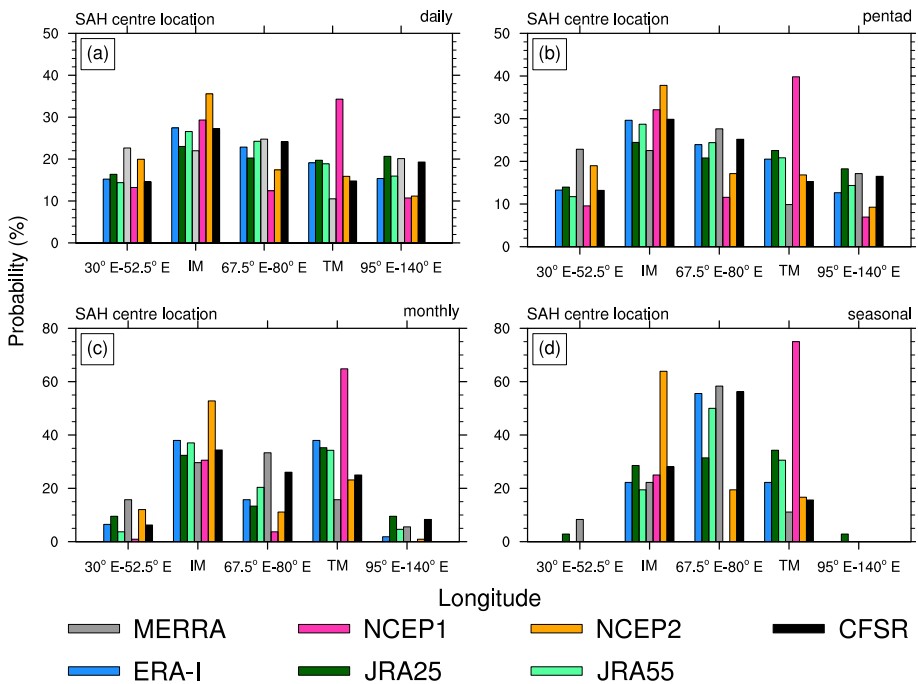

**Figure 2.** Overview of the probability distributions of the SAH's longitudinal location at 100 hPa based on **a)** daily , **b)** pentad, **c)** monthly and **d)** seasonal data during June to August 1979–2014 (1979–2010/1979–2013 for CFSR/JRA25). The IM and TM regions comprise the longitudes 55° E–65° E and 82.5° E–92.5° E, respectively.

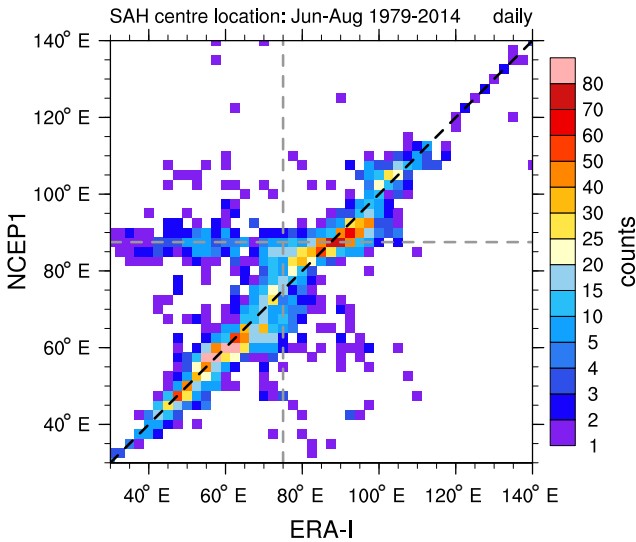

**Figure 3.** Scatter plot of the daily longitudinal location of the SAH centre during June–August 1979–2014 at 100 hPa as diagnosed by ERA-I and NCEP1 (note the nonlinear colour scale). Dashed black line indicates the perfect one-to-one correspondence and grey lines indicate strong deviations from the perfect fit.

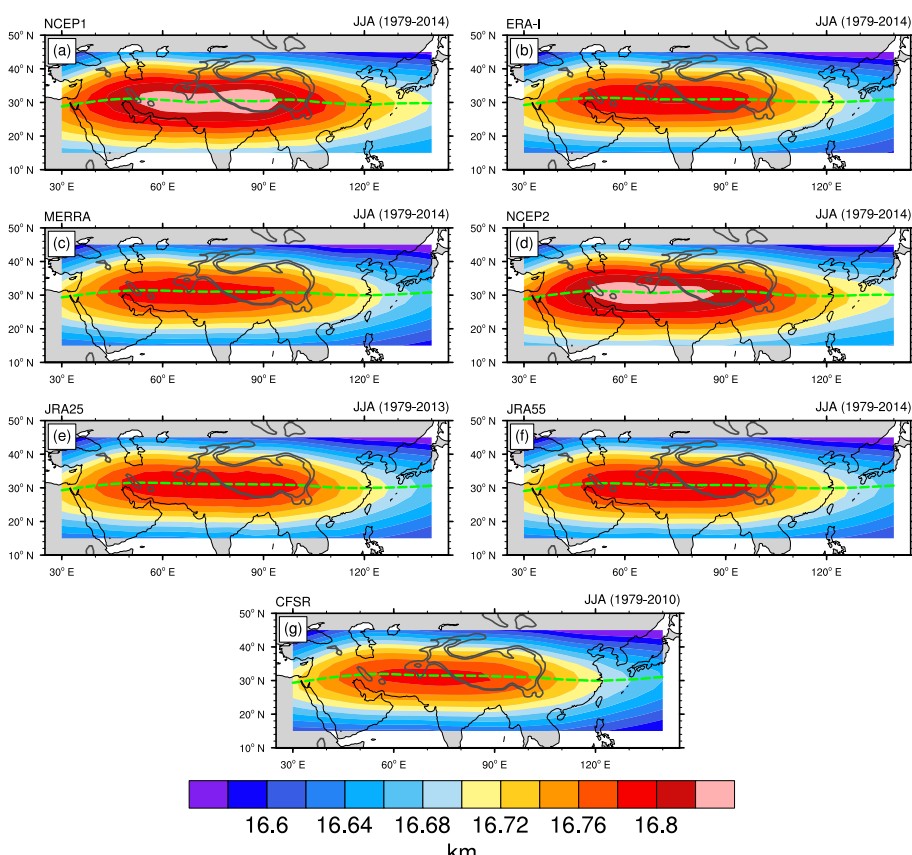

**Figure 4.** Climatology of geopotential height (km) at 100 hPa for the seven reanalysis during JJA 1979–2014. For JRA25/CFSR the period 1979–2013/1979–2010 is considered. Green lines indicate the climatological ridge lines from the data sets. Black contours display the orography of 2 km and 3 km for orientation.

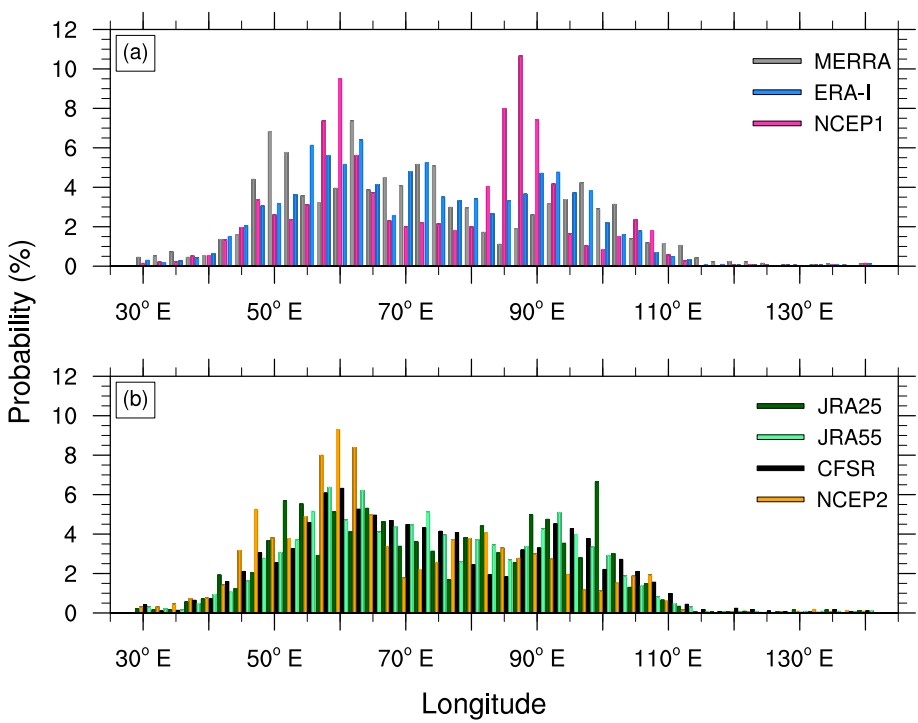

**Figure 5.** PDF of the daily location of the SAH centre at 100 hPa during JJA 1979–2014 (1979–2010/1979–2013 for CFSR/JRA25) for the seven reanalysis. The binning was performed according to the 2.5° resolution of the data.

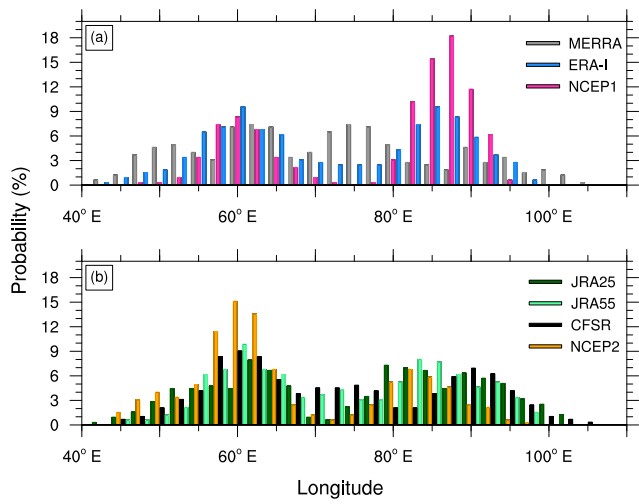

**Figure 6.** As Fig. 5 but for monthly mean data. Additionally the original distribution has been smoothed using a running average over three grid points (7.5°).

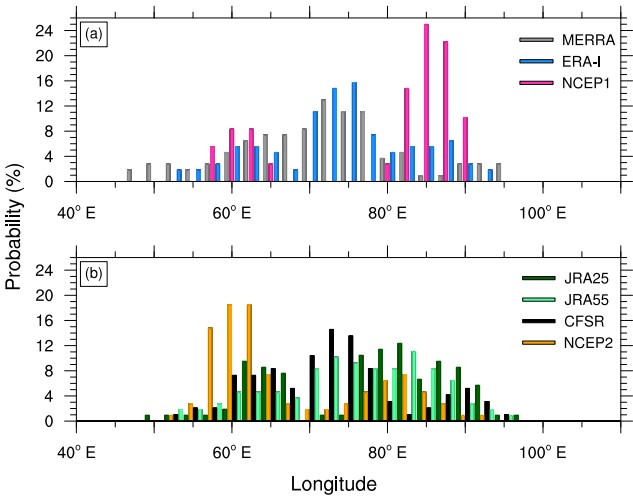

**Figure 7.** As Fig. 6 but for seasonal mean (JJA) data.

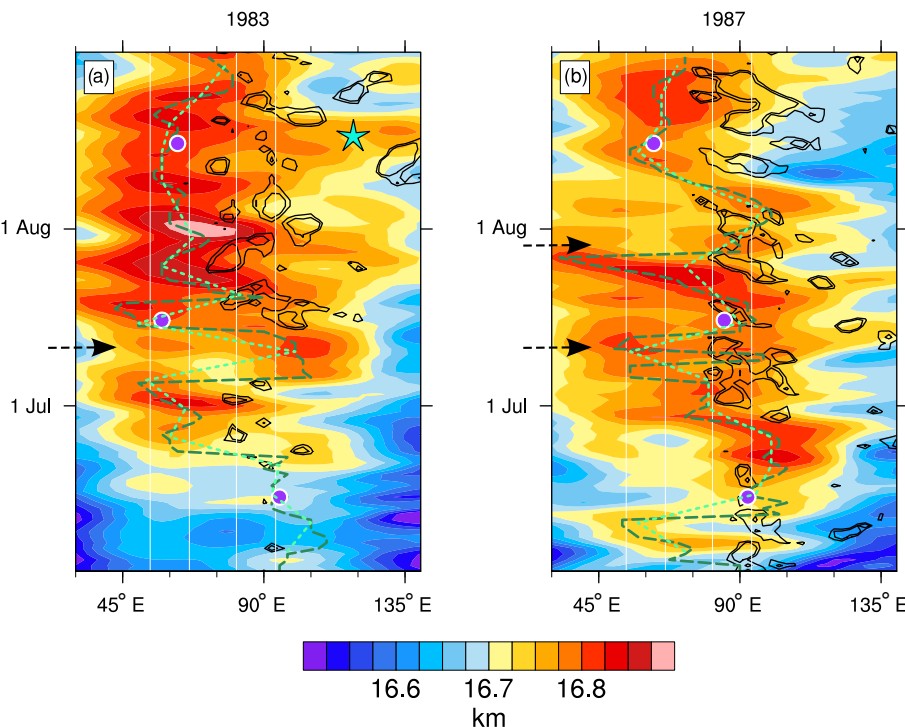

**Figure 8.** Hovmoeller plots of geopotential height (averaged over 20° N–40° N) in km from ERA-I in summer (JJA) **(a)** 1983 and **(b)** 1987. Black contours show OLR (averaged over 15° N–30° N) from NOAA at levels of 180 W m$^{-2}$ (inner contours) and 190 W m$^{-2}$. Dashed dark green and light green show the movement of the SAH centre based on daily and pentad data. Purple dots show the location of the SAH based on monthly data. White lines indicate the IM (left) and TM (right) region. Arrows indicate splitting events discussed in the text. Turquoise star shows the strong shedding event in August 1983.

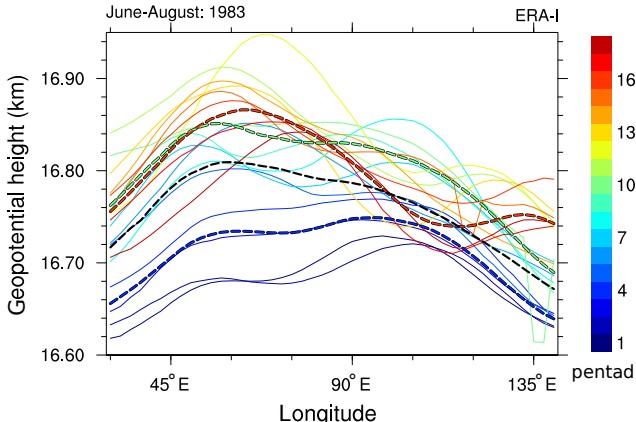

**Figure 9.** Geopotential height (km) along the ridge line of the SAH for individual pentads (colour coded solid lines) in 1983. The first pentad is 3–7 June 1983. Dashed lines show geopotential height along the ridge line based on June (blue), July (green), August (red) and the seasonal (JJA) mean data (black) in 1983.

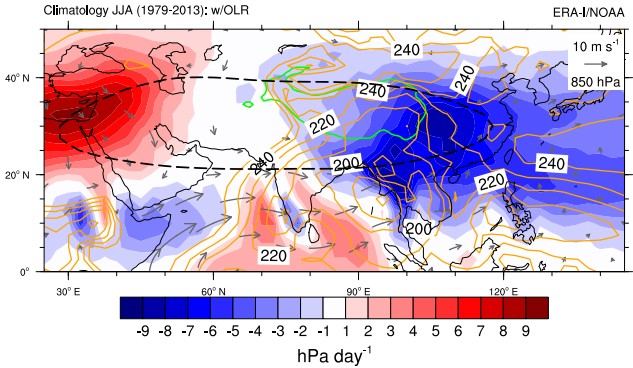

**Figure 10.** Mean vertical velocity (hPa day$^{-1}$) at 100 hPa during JJA (1979–2013) as diagnosed from ERA-I. Upward (downward) winds are indicated by blue (red) colours. Orange contours show mean OLR (W m$^{-2}$) starting from 250 W m$^{-2}$ to 180 W m$^{-2}$ in steps of 10 W m$^{-2}$ during the same period. Black dashed contour indicates the climatological mean position of the SAH (16.72 km contour of geopotential height at 100 hPa). Arrows show the mean JJA (1979–2013) 850 hPa horizontal winds from ERA-I. Green contour indicates the 3 km contour of the Tibetan Plateau.

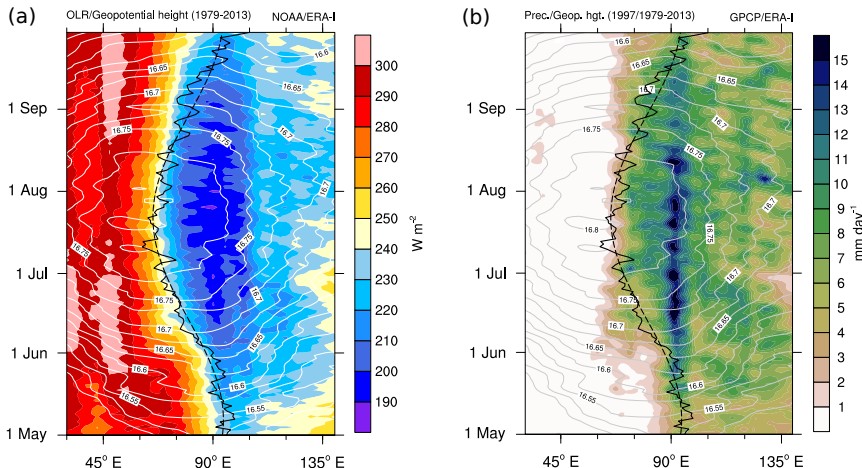

**Figure 11. (a)** Hovmoeller diagram (longitude vs. time) of OLR (NOAA, 15° N–30° N average in W m$^{-2}$) and geopotential height (km) at 100 hPa (ERA-I, 20° N–40° N average, white contours) averaged over the years 1979–2013. The black solid line indicates mean location of the SAH centre during the same period. The dashed line represents the lowpass-filtered mean location of the SAH, i.e. periods less than 80 days are removed from the black solid line. **(b)** as in **(a)** but colour coded is GPCP daily precipitation (mm day$^{-1}$) at 1x1 degree resolution averaged over 15° N–30° N during the years 1997–2013. Precipitation data has been smoothed by 3 degrees longitudinally and 3 days temporally.

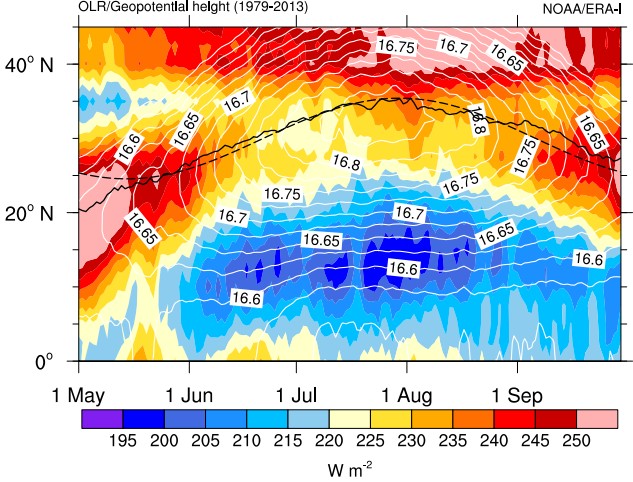

**Figure 12.** As Fig. 11 but for latitude vs. time. OLR and geopotential height (km) at 100 hPa have been averaged over 70° E–130° E and 45° E–100° E, respectively.

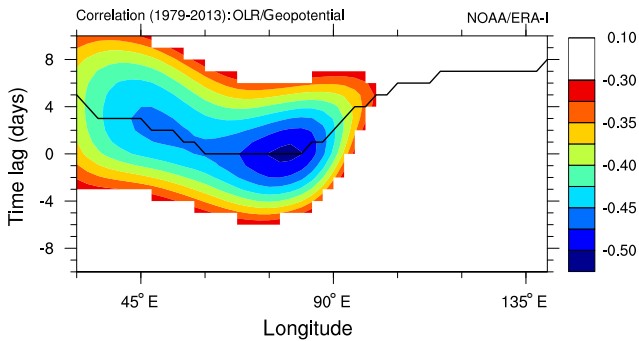

**Figure 13.** Correlation between NOAA-OLR (avgeraged over 15° N–30° N x 70° E–130° E) and ERA-I geopotential (averaged over 20° N–40° N) at 100 hPa depending on time lag and longitude. Positive lags indicate that geopotential change occurs after OLR changes. Correlation is calculated based on the deseasonalised and smoothed (periods shorter than 10 days removed) time series during May to September for every year separately and then averaged over the 35 year period 1979–2013 (period covered by OLR and geopotential data). The black line shows the maximum anticorrelation at each longitude.

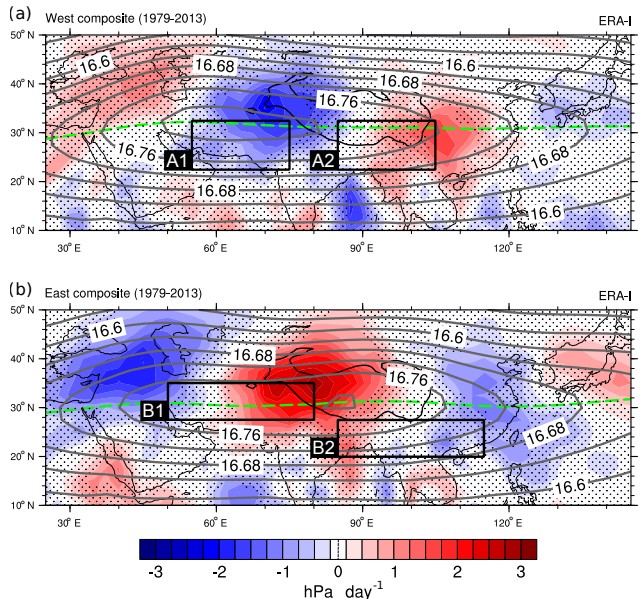

**Figure 14.** Colour shading displays the mean anomaly of the vertical velocity (hPa day$^{-1}$) at 100 hPa from ERA-I during **a)** west and **b)** east summers with respect to the JJA mean during 1979–2013. Stippling indicates where the anomalies are insignificant (significance less than 10%). Grey contours show composites of ERA-I geopotential height at 100hPa (in km) during **a)** west and **b)** east summers. The green lines indicate the ridge lines. Black contour shows orography greater than 3 km. Black boxes show the averaging regions of geopotential, needed to calculate the SAHI$_{14}$ and the SAHI$_{15}$. The SAHI$_{14}$/SAHI$_{15}$ is defined as the standardized difference of average geopotential in box A2/B2 (22.5° N–32.5° N, 85° E–105° E/20° N–27.5° N, 85° E–115° E) minus average geopotential in box A1/B1 (22.5° N–32.5° N, 55° E–75° E/27.5° N–35° N, 50° E–80° E).

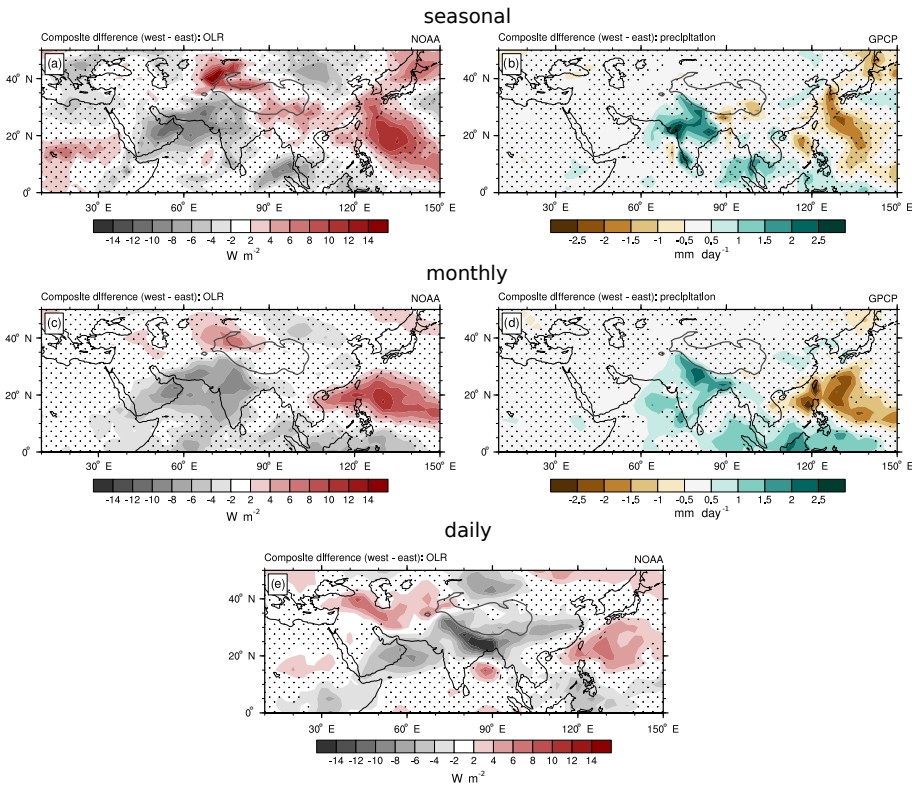

**Figure 15.** Composite differences of west minus east location of the SAH centre in the period 1979–2013 as diagnosed by ERA-I for **(a,c,e)** OLR (from NOAA in W m$^{-2}$) and **(b,d)** precipitation (from GPCP in mm day$^{-1}$) based on **(a-b)** seasonal, **(c-d)** monthly and **(e)** daily data. Stippling indicates insignificant areas (significance less than 10%). Black contours show the location of the TP (orography higher than 3 km).

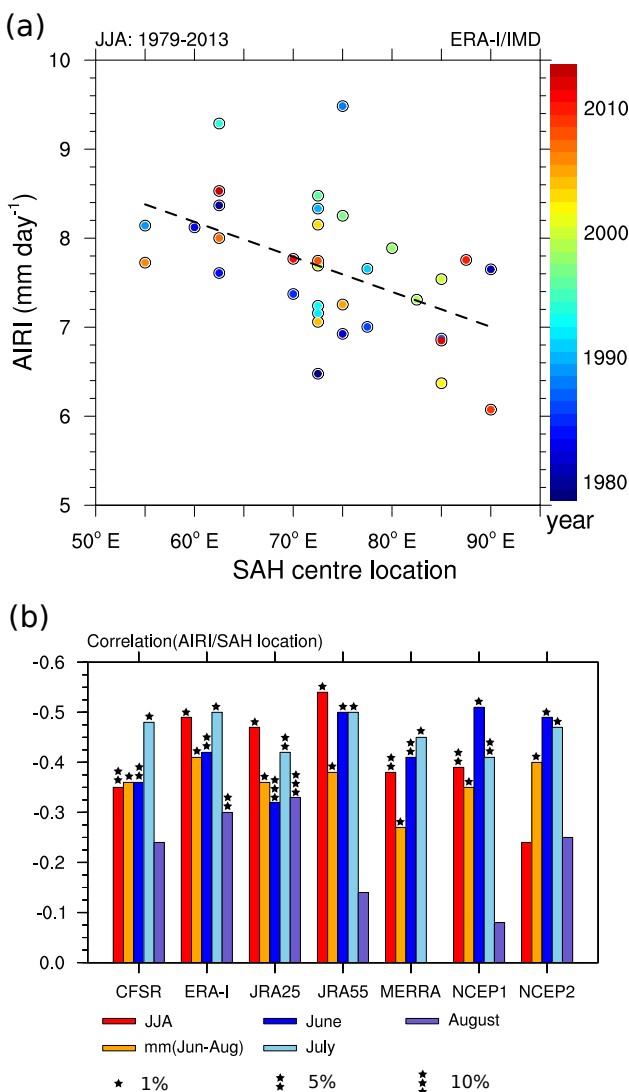

**Figure 16.** **(a)** Scatter plot of AIRI and the SAH location from ERA-I based on JJA data during 1979–2013 (years are colour coded). Black line indicates the regression line. **(b)** Correlation of AIRI with the longitudinal location of the SAH centre based on seasonal (red columns) and monthly mean (orange columns) data during June to August 1979–2013 (1979–2010 for CFSR) for the seven reanalyses. For monthly mean data the time series has been deseasonalised (i.e. multiannual June/July/August values have been subtracted). Blueish colours indicate the correlation for the months June, July and August separately. Stars indicate the respective significance level.

**Table 1.** Overview of the reanalysis data employed in this study.

|  | CFSR | ERA-I | JRA25 | JRA55 |
|---|---|---|---|---|
| reference | Saha et al. (2010) | Dee et al. (2011) | Onogi et al. (2007) | Kobayashi et al. (2015) |
| institution | NCEP | ECMWF | JMA/CRIEPI | JMA |
| resolution |  |  |  |  |
| *horizontal* | T382* ($\sim$0.5° x 0.5°) | T255 ($\sim$0.7° x 0.7°) | T106 ($\sim$1.1° x 1.1°) | TL319** ($\sim$55 km) |
| *vertical (top)* | L64*** ($\sim$0.3hPa) | L60 (0.1hPa) | L40 (0.4hPa) | L60 (0.1hPa) |
| assimilation | 3D–Var | 4D–Var | 3D–Var | 4D–Var |
| data source | rda.ucar.edu | apps.ecmwf.int | rda.ucar.edu | rda.ucar.edu |
| period | Jan 1979–Dec 2010 | Jan 1979–present | Jan 1979–Jan 2014 | Jan 1958–present |

|  | MERRA | NCEP1 | NCEP2 |
|---|---|---|---|
| reference | Rienecker et al. (2011) | Kalnay et al. (1996) | Kanamitsu et al. (2002) |
| institution | NASA | NCEP/NCAR | NCEP/DOE |
| resolution |  |  |  |
| *horizontal* | 0.5° x 0.67° | T62 ($\sim$1.9° x 1.9°) | T62 ($\sim$1.9° x 1.9°) |
| *vertical (top)* | L72 (0.01hPa) | L28 (3hPa) | L28 (3hPa) |
| assimilation | 3D–Var | 3D–Var | 3D–Var |
| data source | mirador.gsfc.nasa.gov | esrl.noaa.gov | rda.ucar.edu |
| period | Jan 1979–present | Jan 1948–present | Jan 1979–present |

* Triangular truncation (T); ** Triangular truncation with linear reduced Gaussian grid (TL); *** Vertical levels (L)

**Table 2.** Overview of studies addressing bimodality. Geopotential height and zonal wind are denoted by $\phi$ and $u$, respectively. For details see Sect. 2.2.

| Study | Data set | Period | Time step | Variables | Method |
|---|---|---|---|---|---|
| Zhang et al. (2002) | NCEP1 | July–August 1980–1994 | pentad | $\phi, u$ at 100 hPa | max $\phi$ along ridge line |
| Qian et al. (2002)* | NCEP1 | July–August 1958–1997 | monthly | $\phi, u$ at 100 hPa | max $\phi$ along ridge line |
| Zhou et al. (2006)* | NCEP1 | June–August 1950–1999 | monthly | $\phi$ at 100 hPa | max $\phi$ |
| Zhou et al. (2009)* | NCEP1 | July–August 1950–1999 | monthly | $\phi$ at 100 hPa | max $\phi$** |
| Zarrin et al. (2010)* | NCEP1 | June–August 1971–2000 | monthly | $\phi$ at 100/200 hPa | max $\phi$*** |
| Yan et al. (2011) | NCEP1 | June–August 2005–2009 | daily | $\phi, u$ at 100 hPa | max $\phi$ along ridge line |
| Wei et al. (2014) | ERA40 | June–August 1958–2002 | seasonal | $\phi, u$ at 200 hPa | max $\phi$ along ridge line** |
| Garny and Randel (2013) | MERRA | May–September 2006 | daily | PV at 360/380 K | probability via PV threshold |
| Ploeger et al. (2015) | ERA-I | 20 June–20 August 2011 | daily | PV at 360/380 K | probability via PV threshold |

* Time periods and height levels have been restricted to the range where bimodality of the SAH has been found in the respective studies. ** Method of centre detection was not specified. The methods in Zhou et al. (2009) and Wei et al. (2014) are probably based on the methods in Zhou et al. (2006) and Zhang et al. (2002), respectively. *** Possibility to detect multiple centres. The study, however, does not focus on the Monsoon region but investigates anticyclones worldwide.

**Table 3.** Correlation of longitudinal and latitudinal location of the SAH centre based on seasonal mean (JJA mean) and monthly mean (deseasonalised) data from June to August 1979–2014 (1979–2010/2013 for CFSR/JRA25). Asterisks indicate the significance levels of the correlation coefficients.

|        | seasonal mean | monthly mean |
|--------|:-------------:|:------------:|
| CFSR   | -0.41**       | -0.37*       |
| ERA-I  | -0.28         | -0.29*       |
| JRA25  | -0.53*        | -0.53*       |
| JRA55  | -0.36**       | -0.35*       |
| MERRA  | -0.48*        | -0.26*       |
| NCEP1  | -0.15         | -0.15        |
| NCEP2  | -0.22         | -0.35*       |

Significance level: ***(0.1), **(0.05), *(0.01).