# Peer review of "Movement, drivers and bimodality of the South Asian High"

_Atmospheric Chemistry and Physics, 2016_

## Referee Comment (RC1) · Anonymous Referee #1 · 11 Jul 2016

This paper analyzes the northern-summer South Asian High (SAH) at 100 hPa and related variables in six reanalysis data sets, and found that the "bimodality" of the SAH is only significant in NCEP1 (and with a lesser extent in NCEP2). All more recent reanalyses do not show such a strong bimodality. This is a very interesting and important work. However, to me, it would be much more appropriate to hypothesize that NCEP1 and NCEP2, the old 1990s reanalysis systems, are wrong in terms of the possible SAH bimodality and to discuss why they are wrong. The authors, however, do not give their answer (or hypothesis) clearly to the question, and continue to make further data analyses. Thus, after reading through the manuscript, I am somewhat at a loss regarding the question in the manuscript title.

It is unfortunate that the authors do not analyze the latest NCEP reanalysis, the CFSR (Saha et al., BAMS, 2010). If the CFSR also shows similar tendencies to the mod-

ern reanalyses, the authors would also be able to obtain advice from the NCEP colleagues why the old NCEP reanalyses are wrong regarding the 100 hPa geopotential height over the middle to south Asian regions. Some hints might be as follows. (1) NCEP1 and NCEP2 are the only reanalyses available now that assimilate temperature retrievals for TOVS and ATOVS measurements; all more recent reanalyses assimilate radiance data directly. (2) NCEP1 and NCEP2 are the only reanalyses that use the sigma coordinate, while all others use the hybrid sigma-pressure coordinate. (3) As discussed by Kanamitsu et al. (2002), NCEP2 is an updated version of NCEP1, where several errors in the original NCEP1 system were corrected; thus, NCEP2 is generally closer to the truth than NCEP1.

Also, it may be useful to make a separate analysis using data only for the recent 10 years when several advanced satellite measurements are available (which NCEP1 and NCEP2 cannot utilize because of the oldness of their system). (Also, though it may not be a key here, investigation on GNSS Radio Occultation impact may also be interesting. MERRA does not assimilate GNSS RO, while MERRA-2, JRA-55, CFSR, and ERA-I do assimilate it.) If there is an influence of changing observing systems, the results of such an analysis would give us another hint.

In Introduction, and in other places, the authors cite several previous works that discuss the SAH's bimodality. It would be important and useful to summarize the information (in a table) on: (1) data used, (2) period of the analysis, (3) variables used, and (4) details of how to define the SAH centre(s) for the SAH bimodality study in each paper. Are there any works that use a reanalysis other than NCEP1?

In the latter half of Section 4 (page 8, lines 34-), and beyond (to the end), I cannot follow the discussion fully because the authors do not give any conclusion which is correct, NCEP1 or ERA-I (and others) regarding the SAH bimodality, and because they switch the main data set to ERA-I and continue discussion.

In conclusion, I think that the SAH's possible bimodality problem is a very interesting and important one, but the current manuscript is not conclusive. Major revisions explained above are necessary.

---

## Referee Comment (RC2) · Anonymous Referee #3 · 13 Sep 2016

The paper presents an analysis of the location of the South Asian High (SAH) at 100 hPa for the time period from 1979 until 2014. In previous studies a bimodality in the longitudinal location of the South Asian High was found in two preferred regions classified into the Tibetan Mode (TM) and the Iranian Mode (IM), respectively. In this study, the existence of a bimodality of the SAH is analyzed in six different reanalyses: (1) The NCEP/NCAR Reanalysis 1 (NCEP1) from the National Centers of Environmental Prediction (NCEP) and the National Center of Atmospheric Research (NCAR), (2) NCEP/DOE Reanalysis 2 (NCEP2) from NCEP and the Department of Energy (DOE), (3) the Japanese 25-year reanalysis (JRA25) from the Japan Meteorological Agency (JMA) and the Central Research Institute of Electric Power Industry (CRIEPI), (4) the 55-year reanalysis (JRA55) from JMA, (5) the ERA-Interim reanalysis (ERA-I) from the European Centre of Medium-range Weather Forecast (ECMWF) and (6) the Modern Era Retrospective-Analysis (MERRA) from the National Aeronautics and Space Administration (NASA) on interannual, seasonal and synoptic time scales.

[Figure]

The main conclusion of the study is that only the NCEP1 reanalysis shows are clear bimodal structure of the longitudinal location of the SAH for daily and 5-day mean data. Moreover, the study demonstrate a connection between westward/eastward movement and northward/southward movement of the SAH based on ERA-Interim reanalysis data. The connection between the location of the SAH, OLR, and rainfall over India and the West Pacific in analyzed.

**General comments:**

The bimodality of the SAH, its drivers and its connection to precipitation is very important and interesting. I recommend that the paper is suitable for publication after revision to address the comments listed below.

The paper is well-structured and significant for publication by ACP. However, section 4 and 5 are not well included within the abstract, introduction and the title (see comments below) and look like an appendix to section 3. For both topics (Sect. 3) and (Sect 4-5), the authors could go deeper into the details (see comments below). The authors might think of splitting this paper into two parts, however this is fully at the decision of the authors.

I believe that the paper will benefit from a bit more discussion of its results in context with previous studies to better demonstrate what is new in this study. I recommend to do that in an separate discussion section. Further, I believe that the paper will benefit from shorter conclusion section highlighting the main results. Some of the discussion could be shifted in the separate discussion section.

**Specific comments:**

I recommend that authors think about a more comprehensive title of the paper. The current title 'Is there bimodality of the South Asian High?' includes only the first part of the work (question 1). Section 4 and 5 (question 2 and 3) are not considered in the current title.

p. 1, line 22-25: It would be helpful for the reader to add here in the introduction briefly how the SAH center location is defined/calculated by Zhang et al. (2002) (e.g. estimated as geopotential height maximum). The method by Zhang et al. (2002) is described in detail in Sec. 2.2, nevertheless I think it is helpful to also describe the basic idea within the introduction.

p. 2, line 18-19: 'Tackling these questions is also subject of past and upcoming measurement campaigns such as ESMVal (2012), OMO (2015) and StratoClim (2016).' This sentence seems a bit out of context here. Perhaps the authors could emphasize here that only very sparse aircraft measurements in altitudes of the SAH were available in the region of the Asian monsoon until now... Further the authors could cite some results of papers that were already published to these (previous) campaigns (e.g. ESMVal/TACTS/OMO) addressing the Asian monsoon anticyclone and its impact on the stratosphere or cite overview papers instead of citing project web pages.

p. 3, line 31: 'Second, along this ridge line the maximum of the daily (pentad/monthly/seasonal) geopotential field at 100 hPa is determined.' My question is here, in case you would have simultaneously two maxima in geopotential field (e.g. splittings of the anticyclone), using this method you would only count the strongest maximum. The impact of the 'second lower' maximum is not taken into account by this method. Is that correct? If yes, the used method would not reflect the full variability of the Asian monsoon anticyclone.

p. 4, line 12: 'As in the studies mentioned before, we choose the 100 hPa level in our study to be consistent with these previous works.' The authors cite before papers by Qian et al. (2002), Zhang et al. (2002), Yan et al. (2011) using the 100 hPa level. In the introduction the authors mention some papers demonstrating the strong variability in strength and location of the SAH: Hsu and Plumb, 2000; Popovic and Plumb, 2001; Garny and Randel, 2013; Ploeger et al., 2015; Vogel et al., 2015. To my knowledge all these studies use levels of potential temperature for their analysis. Thus more studies are mentioned witin the paper using level of potential temperature.

The authors do a good job of analyzing the bimodality using six different reanalysis data including daily, pentad, monthly, and seasonal geopotential fields taking into account the high temporal variability of the location of the SAH and differences between the used data set. However, would the same analysis on a fixed level of potential temperature (e.g. at 370 K or 380 K) result in the same conclusions as for the 100 hPa level? If yes, the result would strengthen the results of the paper. If not, it would be interesting to discuss the differences. I suggest to include in the paper the same analysis, but on a level of potential temperature in addition to the 100 hPa pressure level.

p. 7, line 4-5: 'Common to all reanalyses is that there is a shift of the distribution to the west from June to July and a shift back to the east from July to August.' Is this shift also found in monsoon rainfall patterns?

p. 7, line 7: 'Based on seasonal mean (JJA-mean) data the SAH shows a bimodal structure in the reanalyses NCEP1, NCEP2 and JRA25 (see Fig. 8). Here, NCEP1 and JRA25 show two pronounced peaks over the TP and IP... ' I can't see a bimodal structure in JRA25 in Fig. 8. Do you mean here NCEP2 data?

p 8, line 6: 'To the right we can observe exactly the opposite behavior, i.e. propagation to the east.' Please clarify what does that mean?

p 8, line 24: '..e.g. identifying the Somali-Jet, which brings moisture from the Arabian Sea to India' Please add here a citation.

p 8, line 21: 'For seasonal mean and deseasonalised monthly mean data all reanalyses show that westward (eastward) movement of the SAH is related to northward (southward) movement. The separate analysis of June, July and August yields that this relationship is strong during June and July'. Does that mean that during August the SAH is shifted to the north and convection areas are further south and can not trigger an east-west shift? Please clarify that.

p. 8 (sect. 4) The east-west and north-south movement (shift) of the SAH is also found in previous studies. The authors should add here some references and discuss their results in the context of previous studies. In addition, an east-west and north-south shift is also found in rainfall patterns over India/Asia. I think it would strengthen the findings of the paper, if the authors would discuss this connection. Further, convection and rainfall in this region is strongly impacted by El Nino and La Nina events. Have El Nino events an impact on the correlation between westward (eastward) movement and northward (southward) movement of the SAH?

p. 10, line 7-15: Again is there a connection between east and west phase to the El Nino/Southern Oscillation (ENSO)?

[Figure]

**Technical corrections:**

p.1, line 10: shortcut 'ORL' is used witin the abstract without explanation

p.2, line 11/13: 'downloaded' $->$ 'used' ?

p.4, line 21: 'long term' $->$ 'long-term'

p.6, line 30: remove line break

Figure 1: 'The grey box'.. I assume the authors mean the box marked by the dashed dotted line. If not, please clarify.

---

## Author Comment (AC1) · 19 Oct 2016

**Author Comment to Referee #1**

We thank referee#1 for his helpful comments on the manuscript. The referee's comments are presented in *italics* and our point-by-point answers are highlighted in blue.

- *This paper analyzes the northern-summer South Asian High (SAH) at 100 hPa and related variables in six reanalysis data sets, and found that the "bimodality" of the SAH is only significant in NCEP1 (and with a lesser extent in NCEP2). All more recent reanalyses do not show such a strong bimodality. This is a very interesting and important work. However, to me, it would be much more appropriate to hypothesize that NCEP1 and NCEP2, the old 1990s reanalysis systems, are wrong in terms of the possible SAH bimodality and to discuss why they are wrong. The authors, however, do not give their answer (or hypothesis) clearly to the question, and continue to make further data analyses. Thus, after reading through the manuscript, I am somewhat at a loss regarding the question in the manuscript title.*

  We thank referee#1 for the encouraging classification of the manuscript. Regarding the question of which reanalysis is right or wrong, we can not give a definite answer (allthough the natural reflex would be that the newer "and thus better" reanalyses should be right) and the title was supposed to be in a way rhetorical. In response to referee#3's comment we changed the manuscript title to: **"Movement, drivers and bimodality of the South Asian High"** to better incorporate the whole manuscript. Also, especially in the new Conclusion section we address implications of this result and in the Discussion section we touch how difficult it is to assess which reanalysis is wrong and why. For example, already NCEP1 and NCEP2 show huge differences of the PDFs, allthough the base model and assimilation data is in principle the same, except for the changes mentioned in Kanamitsu et al. (2002). Nevertheless, we address possible reasons for the difference of NCEP1 data in the new discussion section and stress that it is likely that the bimodality is an artefact of NCEP1.

- *It is unfortunate that the authors do not analyze the latest NCEP reanalysis, the CFSR (Saha et al., BAMS, 2010). If the CFSR also shows similar tendencies to the modern reanalyses, the authors would also be able to obtain advice from the NCEP colleagues why the old NCEP reanalyses are wrong regarding the 100 hPa geopotential height over the middle to south Asian regions. Some hints might be as follows.*
  *(1) NCEP1 and NCEP2 are the only reanalyses available now that assimilate temperature retrievals for TOVS and ATOVS measurements; all more recent reanalyses assimilate radiance data directly.*
  *(2) NCEP1 and NCEP2 are the only reanalyses that use the sigma coordinate, while all others use the hybrid sigma-pressure coordinate.*

*(3) As discussed by Kanamitsu et al. (2002), NCEP2 is an updated version of NCEP1, where several errors in the original NCEP1 system were corrected; thus, NCEP2 is generally closer to the truth than NCEP1.*

We thank the reviewer for this helpful comment. Consequently, we have included CFSR data in our study. Regarding the placement of the SAH centre this data set is mostly in agreement with ERA-I and JRA55. Regarding the possiblity to get advice from NCEP colleagues: Our motivation was to spread the word about this discrepancy between NCEP1 and the other reanalyses, because a couple of studies have been referring to the concept of bimodality. We hope that our motivation and the implications are more clearly stated now in the separate Conclusion (Sect. 7). Still we included the hints (1) and (2) in the Discussion (Sect. 6). Hint (3) has already been mentioned in the first manuscript version (maybe a little to hidden/detached) and was hence shifted to the Discussion (Sect. 6) as well.

- *Also, it may be useful to make a separate analysis using data only for the recent 10 years when several advanced satellite measurements are available (which NCEP1 and NCEP2 cannot utilize because of the oldness of their system). (Also, though it may not be a key here, investigation on GNSS Radio Occultation impact may also be interesting. MERRA does not assimilate GNSS RO, while MERRA-2, JRA-55, CFSR, and ERA-I do assimilate it.) If there is an influence of changing observing systems, the results of such an analysis would give us another hint.*

  The results for daily data have been analysed for the recent 10 years without significant changes compared to the full time period. We mention this analysis in the Discussion (Sect. 6).

- *In Introduction, and in other places, the authors cite several previous works that discuss the SAHs bimodality. It would be important and useful to summarize the information (in a table) on: (1) data used, (2) period of the analysis, (3) variables used, and (4) details of how to define the SAH centre(s) for the SAH bimodality study in each paper. Are there any works that use a reanalysis other than NCEP1?*

  As suggested by the referee, we have added a table summarising this information (Table 2 in the revised manuscript). To our knowledge the study by Wei et al. (2014) is the only study on pressure levels not working with NCEP1 data that shows a clear bimodality. Wei et al. (2014) use ERA40 data at 200 hPa, as does their follow up study (Wei et al., 2015). We did not include ERA40 in our study as this reanalysis is only available until 2002.

- *In the latter half of Section 4 (page 8, lines 34-), and beyond (to the end), I cannot follow the discussion fully because the authors do not give any conclusion which is correct, NCEP1 or ERA-I (and others) regarding the*

*SAH bimodality, and because they switch the main data set to ERA-I and continue discussion.*

We agree with the referee, however, as said before we cannot claim that NCEP1 is false, as we do not have certain proof of this fact. Hence we expanded the transition from Section 3 to Section 4 which now reads: **"The salient disagreement of the reanalyses in the distribution of the SAH center location is our motivation to revisit the questions of how the SAH moves on various time scales and how this movement is caused. To tackle these questions, we will focus on results based on observational and ERA-I data during the next two sections (Sects. 4 and 5). We choose ERA-I as it is a heavily used reanalysis with the most recent data assimilation scheme. Apart from that, our choice is arbitrary and we address the sensitivity of the presented results with respect to the reanalysis in the discussion (Sect. 6)."** Before, the discussion of the sensitivities has been spread between the respective Sections and the previous Section 6 "Discussion and Summary". We hope that our line of argument is easier to follow now.

- *In conclusion, I think that the SAHs possible bimodality problem is a very interesting and important one, but the current manuscript is not conclusive. Major revisions explained above are necessary.*

  Again we thank the reviewer for this positive feedback and hope that our reply and the revised version of the manuscript answer the points raised by the referee.

**References**

M. Kanamitsu, W. Ebisuzaki, J. Woollen, S.-K. Yang, J. J. Hnilo, M. Fiorino, and G. L. Potter. NCEP-DOE AMIP-II Reanalysis (R-2). *B. Am. Meteorol. Soc.*, 83(11):1631–1643, Nov. 2002. ISSN 0003-0007. doi: 10.1175/ BAMS-83-11-1631. URL http://dx.doi.org/10.1175/BAMS-83-11-1631.

W. Wei, R. Zhang, M. Wen, X. Rong, and T. Li. Impact of Indian summer monsoon on the South Asian High and its influence on summer rainfall over China. *Clim. Dynam.*, 43(5-6):1257–1269, 2014. ISSN 0930-7575. doi: 10.1007/s00382-013-1938-y. URL http://dx.doi.org/10.1007/ s00382-013-1938-y.

W. Wei, R. Zhang, M. Wen, B.-J. Kim, and J.-C. Nam. Interannual Variation of the South Asian High and Its Relation with Indian and East Asian Summer Monsoon Rainfall. *J. Climate*, 28(7):2623–2634, Dec. 2015. ISSN 0894-8755. doi: 10.1175/JCLI-D-14-00454.1. URL http://dx.doi.org/10. 1175/JCLI-D-14-00454.1.

---

## Author Comment (AC2) · 19 Oct 2016

**Author Comment to Referee #3**

We thank referee#3 for his helpful comments on the manuscript. The referee's comments are presented in *italics* and our point-by-point answers are highlighted in blue.

*The paper presents an analysis of the location of the South Asian High (SAH) at 100 hPa for the time period from 1979 until 2014. In previous studies a bimodality in the longitudinal location of the South Asian High was found in two preferred regions classified into the Tibetan Mode (TM) and the Iranian Mode (IM), respectively. In this study, the existence of a bimodality of the SAH is analyzed in six different reanalyses: (1) The NCEP/NCAR Reanalysis 1 (NCEP1) from the National Centers of Environmental Prediction (NCEP) and the National Center of Atmospheric Research (NCAR), (2) NCEP/DOE Reanalysis 2 (NCEP2) from NCEP and the Department of Energy (DOE), (3) the Japanese 25-year reanalysis (JRA25) from the Japan Meteorological Agency (JMA) and the Central Research Institute of Electric Power Industry (CRIEPI), (4) the 55-year reanalysis (JRA55) from JMA, (5) the ERA-Interim reanalysis (ERA-I) from the European Centre of Medium-range Weather Forecast (ECMWF) and (6) the Modern Era Retrospective-Analysis (MERRA) from the National Aeronautics and Space Administration (NASA) on interannual, seasonal and synoptic time scales.*
*The main conclusion of the study is that only the NCEP1 reanalysis shows are clear bimodal structure of the longitudinal location of the SAH for daily and 5-day mean data. Moreover, the study demonstrate a connection between westward/eastward movement and northward/southward movement of the SAH based on ERA-Interim reanalysis data. The connection between the location of the SAH, OLR, and rainfall over India and the West Pacific in analyzed.*

*General comments:*

- *The bimodality of the SAH, its drivers and its connection to precipitation is very important and interesting. I recommend that the paper is suitable for publication after revision to address the comments listed below.*

  We thank referee#3 for this very positive and encouraging comment and the apt summary of our study.

- *The paper is well-structured and significant for publication by ACP. However, section 4 and 5 are not well included within the abstract, introduction and the title (see comments below) and look like an appendix to section 3. For both topics (Sect. 3) and (Sect 4-5), the authors could go deeper into the details (see comments below). The authors might think of splitting this paper into two parts, however this is fully at the decision of the authors.*

  Again we thank the reviewer for this positive feedback and acknowledge, that the title did not incorporate the whole manuscript. Hence we changed the title to: **"Movement, drivers and bimodality of the South**

**Asian High"**. We think that the different sections of the manuscript belong together and hope that in the revised version this connection is presented more clearly. Thus we refrain from splitting the paper into two parts. Of course, the suggested extensions of the analyses have been considered though and are addressed below.

- *I believe that the paper will benefit from a bit more discussion of its results in context with previous studies to better demonstrate what is new in this study. I recommend to do that in an separate discussion section. Further, I believe that the paper will benefit from shorter conclusion section highlighting the main results. Some of the discussion could be shifted in the separate discussion section.*

  We followed the referee's suggestion to split the former Section 6 (Summary and Discussion) into two sections. Section 6 now contains the discussion of our results and section 7 is the Conclusion section. As recommended we shifted additional parts containing discussion elements to the Discussion section. We hope that the additional references given (see comments below) and the restructuring of the manuscript (especially the new Discussion and Conclusion section) will help to contextualise our work.

*Specific comments:*

- *I recommend that authors think about a more comprehensive title of the paper. The current title Is there bimodality of the South Asian High? includes only the first part of the work (question 1). Section 4 and 5 (question 2 and 3) are not considered in the current title.*

  Agreed. As mentioned before, we updated the title to better incorporate the whole manuscript.

- *p. 1, line 22-25: It would be helpful for the reader to add here in the introduction briefly how the SAH center location is defined/calculated by Zhang et al. (2002) (e.g. estimated as geopotential height maximum). The method by Zhang et al. (2002) is described in detail in Sec. 2.2, nevertheless I think it is helpful to also describe the basic idea within the introduction.*

  Done. We changed the sentence referring to Zhang et al. (2002), so the method is stated. It now reads: **"Apart from the highly variable synoptic behaviour of the SAH, Zhang et al. (2002) have found that the longitudinal distribution of the SAH centre location - as identified by the geopotential height maximum along the ridge line (see green line in Fig. 1a) - is bimodal."**

- *p. 2, line 18-19: Tackling these questions is also subject of past and upcoming measurement campaigns such as ESMVal (2012), OMO (2015) and StratoClim (2016). This sentence seems a bit out of context here. Perhaps the authors could emphasize here that only very sparse aircraft*

*measurements in altitudes of the SAH were available in the region of the Asian monsoon until now... Further the authors could cite some results of papers that were already published to these (previous) campaigns (e.g. ESMVal/TACTS/OMO) addressing the Asian monsoon anticyclone and its impact on the stratosphere or cite overview papers instead of citing project web pages.*

We agree with the reviewer and hence we rephrased this part. We now mention studies related to TACTS/ESMVal and the Asian monsoon (Vogel et al., 2014; Müller et al., 2016; Vogel et al., 2016). For the more recent and directly monsoon targeted campaigns OMO and StratoClim (to our knowledge) no studies incorporating measurements (or overview papers) are available so far.

- *p. 3, line 31: Second, along this ridge line the maximum of the daily (pen- tad/monthly/seasonal) geopotential field at 100 hPa is determined. My question is here, in case you would have simultaneously two maxima in geopotential field (e.g. splittings of the anticyclone), using this method you would only count the strongest maximum. The impact of the second lower maximum is not taken into account by this method. Is that correct? If yes, the used method would not reflect the full variability of the Asian monsoon anticyclone.*

This is correct. However, this is the common method to analyse the location of the SAH centre as can be inferred from the new Table 2. As we are referring to these previous studies we chose to keep our analysis consistent. In first tests with an adapted analysis using ERA-I daily data, we allowed for a second maximum (when a clear, spatially separated second local maximum was found). An additional maximum was found in roughly 16% of the days. The qualitative results of our analysis remained unchanged, however.

- *p. 4, line 12: As in the studies mentioned before, we choose the 100 hPa level in our study to be consistent with these previous works. The authors cite before papers by Qian et al. (2002), Zhang et al. (2002), Yan et al. (2011) using the 100 hPa level. In the introduction the authors mention some papers demonstrating the strong variability in strength and location of the SAH: Hsu and Plumb, 2000; Popovic and Plumb, 2001; Garny and Randel, 2013; Ploeger et al., 2015; Vogel et al., 2015. To my knowledge all these studies use levels of potential temperature for their analysis. Thus more studies are mentioned witin the paper using level of potential temperature.*

We agree with the referee's comment. This was a unclear statement in the first manuscript. The sentence "As in the studies mentioned before,..." was meant to refer to the studies mentioned directly ahead of it in the Section 2.2 (Method). These studies explicitly deal with and report bimodality and are all based on pressure levels. This part has been rephrased and we hope that the new Table 2 clarifies our original statement.

- *The authors do a good job of analyzing the bimodality using six different reanalysis data including daily, pentad, monthly, and seasonal geopotential fields taking into account the high temporal variability of the location of the SAH and differences between the used data set. However, would the same analysis on a fixed level of potential temperature (e.g. at 370 K or 380 K) result in the same conclusions as for the 100 hPa level? If yes, the result would strengthen the results of the paper. If not, it would be interesting to discuss the differences. I suggest to include in the paper the same analysis, but on a level of potential temperature in addition to the 100 hPa pressure level.*

  This point is also connected to the comment before regarding anlyses on pressure levels vs. isentropes. We hope that the clarification in the previous comment explains why we are focusing on pressure levels. Nevertheless, we analysed the maximum location of the Montgomery streamfunction on the 395K from ERA-I. (This level is available directly from ECMWF.) Consequently this paragraph has been added to the discussion section: **"Previous studies which address the bimodality of the SAH have mostly focused on the 100 hPa level (see Table 2). To see how robust our results are, we employed ERA-I on the 395 K level. The SAH centre location was defined as the maximum of the Montgomery streamfunction along the ridge line. We found that the PDFs of the SAH centre location with respect to daily and monthly data are similar to ERA-I on 100 hPa. For the seasonal mean data we have found that the distribution changes in favour of the TM and IM, i.e. for seasonal data 12, 10, 14 years are located in the IP, mid and TP region, respectively."**

- *p. 7, line 4-5: Common to all reanalyses is that there is a shift of the distribution to the west from June to July and a shift back to the east from July to August. Is this shift also found in monsoon rainfall patterns?*

  We hope that the inclusion of daily GPCP data in the new Fig. 11 b) and the corresponding paragraph in Sect. 4: **"The seasonal east–west shift can be also found in daily precipitation data from GPCP during the period 1997–2013 (see Fig. 11 b) and the seasonal northward migration of precipitation has been noted in previous studies (e.g. Yihui and Chan, 2005, their Figure 3)."** answers this question. Out of structuring reasons we refrain from stating this at this point in the manuscript, though.

- *p. 7, line 7: Based on seasonal mean (JJA-mean) data the SAH shows a bimodal structure in the reanalyses NCEP1, NCEP2 and JRA25 (see Fig. 8). Here, NCEP1 and JRA25 show two pronounced peaks over the TP and IP... I cant see a bimodal structure in JRA25 in Fig. 8. Do you mean here NCEP2 data?*

  Agreed. We changed this paragraph to: **"Based on seasonal mean (JJA-mean) data the SAH shows a bimodal structure in the re-**

**analyses NCEP1 and NCEP2 (see Fig. 7). Here, NCEP1 shows
a pronounced peak over the TP and a second one over the IP,
whereas NCEP2 shows only a sharp peak over the IP. Addition-
ally, JRA25 shows low probabilities around 70° E. In contrast,
CFSR, ERA-I, JRA55 and MERRA show high probabilities over
the whole centre region (∼60° E–85° E, depending on the reanal-
ysis)."**

- *p 8, line 6: To the right we can observe exactly the opposite behavior, i.e.
  propagation to the east. Please clarify what does that mean?*

  We rephrased this part and hope that our statement is clearer now. The
  paragraph now reads: **"In Fig. 8 the lowest OLR values are mostly
  confined to the area 75° E–105° E and are mostly located east of
  the highest geopotential height values. East of the OLR mini-
  mum we can observe eastward migration of high geopotential,
  associated with eastward eddy shedding of the anticyclone. A
  strong shedding event is observed in mid August 1983 (turquoise
  star in Fig. 8a). West of the OLR minimum region, the core of
  the anticyclone usually propagates westwards."**.

- *p 8, line 24: ..e.g. identifying the Somali-Jet, which brings moisture from
  the Arabian Sea to India Please add here a citation.*

  Done. The reference Rodwell and Hoskins (1995) has been added.

- *p 8, line 21: For seasonal mean and deseasonalised monthly mean data all
  reanalyses show that westward (eastward) movement of the SAH is related
  to northward (southward) movement. The separate analysis of June, July
  and August yields that this relationship is strong during June and July.
  Does that mean that during August the SAH is shifted to the north and
  convection areas are further south and can not trigger an east-west shift?
  Please clarify that.*

  This analysis is simply looking for the general movement without explic-
  itly naming the drivers here. The changes in correlation are indicating
  that the withdrawal is not as clearly towards southeast as the shift to-
  wards northwest during the build up phase of the SAH. The withdrawal
  seems to be first rather eastwards and probably consequently back south-
  wards/equatorwards. This can be inferred from the black solid lines in
  Figs. 11 and 12. By the end of September the longitudinal position is the
  same as at the beginning of May. The southward withdrawal, however,
  seems to take longer than the northward migration during the first part
  of the monsoon season.

- *p. 8 (sect. 4) The east-west and north-south movement (shift) of the SAH
  is also found in previous studies. The authors should add here some refer-
  ences and discuss their results in the context of previous studies. In addi-
  tion, an east-west and north-south shift is also found in rainfall patterns*

*over India/Asia. I think it would strengthen the findings of the paper, if the authors would discuss this connection. Further, convection and rainfall in this region is strongly impacted by El Nino and La Nina events. Have El Nino events an impact on the correlation between westward (eastward) movement and northward (southward) movement of the SAH?*

The shift of the SAH and precipitation is included in the new Fig. 11 b) and mentioned in Sect. 4. The related discussion is also referring to the work of Lau et al. (1988) (which we did not know of during the preparation of the first manuscript) and Yihui and Chan (2005). If referee#3 wants a specific reference to be named, we would be happy to include it in the manuscript. Further, we hope that the discussion of Wei et al. (2014, 2015) is thorough enough in the new discussion section. The connection of ENSO (in terms of Nino 3.4) with the ISM/SAH location was investigated using data from CPC and the corresponding results are now mentioned in the Discussion section (Sect. 6).

- *p. 10, line 7-15: Again is there a connection between east and west phase to the El Nino/Southern Oscillation (ENSO)?*

  Please see comment above.

*Technical corrections:*

- *p.1, line 10: shortcut ORL is used witin the abstract without explanation*

  Done.

- *p.2, line 11/13: downloaded → used ?*

  Done. "Downloaded" is no longer mentioned in the paragraph. The paragraph now reads: **"The data used in this study cover the NH summer seasons 1979 to 2014 (2010/2013 for CFSR/JRA25). Meteorological fields (geopotential height, wind and surface temperature) of all reanalysis data sets have been used with the provided resolution of $2.5° \times 2.5°$, except for MERRA which has been regridded from the native resolution ($0.5°$ latitude by $0.67°$ longitude) to a $2.5° \times 2.5°$ grid..."**

- *p.4, line 21: long term → long-term*

  Done.

- *p.6, line 30: remove line break*

  Done.

- *Figure 1: The grey box.. I assume the authors mean the box marked by the dashed dotted line. If not, please clarify.*

  Done. Following the reviewer's description, this part in the figure caption has been changed to: **"The box marked by the grey dashed dotted line indicates the range of the data which is used to diagnose the centre."**

**References**

K.-M. Lau, G. J. Yang, and S. H. Shen. Seasonal and Intraseasonal Climatology of Summer Monsoon Rainfall over Eeat Asia. *Mon. Weather Rev.*, 116 (1):18–37, 1988. doi: 10.1175/1520-0493(1988)116⟨0018:SAICOS⟩2.0.CO;2. URL `http://dx.doi.org/10.1175/1520-0493(1988)116<0018:SAICOS>2.0.CO;2`.

S. Müller, P. Hoor, H. Bozem, E. Gute, B. Vogel, A. Zahn, H. Bönisch, T. Keber, M. Krämer, C. Rolf, M. Riese, H. Schlager, and A. Engel. Impact of the Asian monsoon on the extratropical lower stratosphere: trace gas observations during TACTS over Europe 2012. *Atmos. Chem. Phys.*, 16(16):10573–10589, 2016. doi: 10.5194/acp-16-10573-2016. URL `http://www.atmos-chem-phys.net/16/10573/2016/`.

M. J. Rodwell and B. J. Hoskins. A Model of the Asian Summer Monsoon. Part II: Cross-Equatorial Flow and PV Behavior. *J. Atmos. Sci.*, 52(9): 1341–1356, 1995. doi: 10.1175/1520-0469(1995)052⟨1341:AMOTAS⟩2.0.CO; 2. URL `http://dx.doi.org/10.1175/1520-0469(1995)052<1341:AMOTAS>2.0.CO;2`.

B. Vogel, G. Günther, R. Müller, J.-U. Grooß, P. Hoor, M. Krämer, S. Müller, A. Zahn, and M. Riese. Fast transport from Southeast Asia boundary layer sources to northern Europe: rapid uplift in typhoons and eastward eddy shedding of the Asian monsoon anticyclone. *Atmos. Chem. Phys.*, 14(23):12745–12762, 2014. doi: 10.5194/acp-14-12745-2014. URL `http://www.atmos-chem-phys.net/14/12745/2014/`.

B. Vogel, G. Günther, R. Müller, J.-U. Grooß, A. Afchine, H. Bozem, P. Hoor, M. Krämer, S. Müller, M. Riese, C. Rolf, N. Spelten, G. P. Stiller, J. Ungermann, and A. Zahn. Long-range transport pathways of tropospheric source gases originating in Asia into the northern lower stratosphere during the Asian monsoon season 2012. *Atmos. Chem. Phys. Disc.*, 2016:1–40, 2016. doi: 10.5194/acp-2016-463. URL `http://www.atmos-chem-phys-discuss.net/acp-2016-463/`.

W. Wei, R. Zhang, M. Wen, X. Rong, and T. Li. Impact of Indian summer monsoon on the South Asian High and its influence on summer rainfall over China. *Clim. Dynam.*, 43(5-6):1257–1269, 2014. ISSN 0930-7575. doi: 10.1007/s00382-013-1938-y. URL `http://dx.doi.org/10.1007/s00382-013-1938-y`.

W. Wei, R. Zhang, M. Wen, B.-J. Kim, and J.-C. Nam. Interannual Variation of the South Asian High and Its Relation with Indian and East Asian Summer Monsoon Rainfall. *J. Climate*, 28(7):2623–2634, Dec. 2015. ISSN 0894-8755. doi: 10.1175/JCLI-D-14-00454.1. URL `http://dx.doi.org/10.1175/JCLI-D-14-00454.1`.

D. Yihui and L. J. C. Chan. The East Asian summer monsoon: an overview. *Meteorol. Atmos. Phys.*, 89(1):117–142, 2005. ISSN 1436-5065. doi: 10.1007/s00703-005-0125-z. URL `http://dx.doi.org/10.1007/s00703-005-0125-z`.

Q. Zhang, G. Wu, and Y. Qian. The Bimodality of the 100 hPa South Asia High and its Relationship to the Climate Anomaly over East Asia in Summer. *J. Meteorol. Soc. Jpn.*, 80(4):733–744, 2002. doi: 10.2151/jmsj.80.733.